

# The significant role of snow in shaping alpine treeline responses in modelled boreal forests

Sarah Haupt[1,2], Josias Gloy[1], Luca Farkas[1], Katharina Schildt[1,2], Lisa Trimborn[1,3], Stefan Kruse[1]

[1]Alfred Wegener Institute, Helmholtz Centre for Polar and Marine Research, 14473 Potsdam, Germany
[2]Institute of Biochemistry and Biology, University of Potsdam, 14476 Potsdam, Germany
[3]Geography Department, Humboldt University Berlin, 10099 Berlin, Germany

*Correspondence to*: Stefan Kruse (stefan.kruse@awi.de); Sarah Haupt (sarah.haupt@awi.de)

**Abstract.** Treelines across the Northern Hemisphere are shifting upward and northward in response to global warming, particularly in boreal forests, where climate change progresses more rapidly at high elevations and latitudes. These shifts

intensify competition for resources, threaten endemic alpine species, and disrupt established ecological relationships, leading to biodiversity loss. However, significant heterogeneity and regional variation exist in how treelines respond to environmental changes, with many underlying drivers and constraints still poorly understood.

This study aims to enhance understanding of alpine treeline dynamics and improve vegetation model predictions under changing climatic conditions. We evaluated the relative impact of key factors influencing treeline migration velocity and

examined the effects of varying snow regimes on treeline migration within the alpine treeline ecotone. To achieve this, we incorporated a novel snow module into the vegetation model LAVESI (*Larix* Vegetation Simulator), enabling the integration of precipitation outside the growing season, snow accumulation, and snowmelt processes. This module allows for explicit modelling of the positive and negative impacts of snow depth on tree growth and treeline migration, while accounting for stochastically occurring extreme events and capturing full weather variability.

Our findings reveal site-specific responses to factors driving treeline shifts and forest expansion, with localised conditions playing a critical role in shaping migration dynamics. The Canadian and the Russian sites demonstrate clear insights into primary migration drivers, while the high variability at the Alaskan site indicates more complex local dynamics and greater predictive uncertainty. The study highlights the significant role of snow in modulating migration potential, as snow accumulation creates favourable conditions for seedling germination and growth while also posing risks of increased mortality

from snow loads or avalanches. These results underscore the importance of incorporating snow-related processes into vegetation models to improve the accuracy of predictions for boreal forest dynamics.

Overall, this study provides valuable insights into tree migration processes, highlighting the varied predictability of treeline responses across regions. These findings carry significant implications for refining vegetation models and guiding conservation strategies to sustain alpine tundra resilience in the face of accelerating climate change.





## 1 Introduction

### 1.1 Importance of treeline research

Treelines across the Northern Hemisphere are shifting upward and northward in response to global warming (Harsch et al., 2009; Holtmeier and Broll, 2010; Lu et al., 2020). This trend is particularly evident in the alpine treelines in the boreal forests, as climate change is progressing more quickly at high elevations and latitudes than in other regions (IPCC, 2022). Several studies report elevational treeline advancement in the mountainous regions of Eurasia, such as the Altai Mountains (Cazzolla Gatti et al., 2019) and Polar Urals (Devi et al., 2008; 2020), and in North America, including the Kenai Mountains (Dial et al., 2007) and Yukon Territory (Danby and Hik, 2007).

The upward expansion of trees into higher elevations intensifies competition for resources such as light, water, nutrients, and habitat availability, threatening endemic alpine species that thrive in open tundra or meadow ecosystems. This competition can limit the growth and survival of native alpine plants adapted to specific environmental conditions now altered by the presence of trees (Greenwood and Jump, 2014). Consequently, treeline advancement may lead to the extinction of specialised flora and fauna unable to adapt or migrate quickly enough (He et al., 2023), resulting in biodiversity loss and the disruption of established ecological relationships (Hansson et al., 2021). Current studies reveal significant heterogeneity and regional variation in the responses of both elevational and northern treelines to changing environmental conditions (Holtmeier and Broll, 2010). Despite ongoing research, the specific drivers and, especially, the constraints affecting how alpine treelines in boreal forests respond to a changing climate are not yet fully understood (He et al., 2023).

### 1.2 Time lag in treeline response to climate warming

Many factors, both favourable and unfavourable, influence the establishment of trees within the treeline ecotone and thus the advance of the treeline. First and foremost, climatic factors determine the position of the treeline. During his research expeditions in the late 18th and early 19th centuries, Alexander von Humboldt observed a pronounced correlation between the latitudinal and elevational treelines and corresponding temperatures. Humboldt integrated this revelation into his pioneering concept of isotherms, delineating lines that connect elevations experiencing equivalent temperatures. This innovative framework allowed him to systematically connect mountains across the globe based on their respective treelines, thereby revolutionising our understanding of climatic patterns and their spatial distribution on a global scale. Körner (1998) introduced the "growth limitation hypothesis" (GLH), which explains cool air temperatures during the growing season as the primary factor determining the upper limits of treelines. Contemporary observations of the GLH indicate that the global treeline follows a Humboldtian isotherm, with mean growing season air temperatures (GSAT) around 6.7°C (Körner and Paulsen, 2004) and 6.4°C (Körner, 2012; Paulsen and Körner, 2014). Treelines warmer than this mean GSAT 6-7°C isotherm (Millar et al., 2020) indicate that tree growth is not solely constrained by cold GSAT and that other factors may play significant roles in determining the position of the treeline (Maher et al., 2021). Numerous studies indicate a significant time lag between climate change progression and the corresponding latitudinal or elevational advance of treelines (Holtmeier, 1985; Hofgaard and Wilmann,



2002; Lloyd, 2005). The fact that only a little more than half of all treelines studied around the world have advanced in response to anthropogenic warming (Harsch et al., 2009) underlines the need to consider factors other than GSAT in vegetation models. The use of vegetation models is crucial for predicting how trees respond to changing climatic conditions. Numerous simulation
studies have been carried out using dynamic global vegetation models (DGVMs), such as SEIB-DGVM (Spatially Explicit Individual Based DGVM) (Smith et al., 2001) or LPJ-GUESS (Lund-Potsdam-Jena General Ecosystem Simulator) (Sato et al., 2007). In this study, we use the individual-based and spatially explicit model LAVESI (*Larix* Vegetation Simulator) (Kruse et al., 2016), which was originally developed to simulate the latitudinal migration of larch forest treelines in northern Siberia. Studies using LAVESI, which already incorporates many of the factors influencing treeline migration rates, also indicate that
vegetation models often overestimate the transition from treeless tundra to forest (Kruse et al., 2016). While temperature is widely recognised as the primary driver of tree growth in this region, dispersal constraints may significantly hinder the northward migration of treelines. These findings underscore the pronounced time lag exhibited by tree populations in response to climate change. However, despite the inclusion of many relevant factors in LAVESI, critical fine-scale processes that are essential for accurately estimating vegetation dynamics may still be overlooked.

**1.3 Factors influencing alpine treeline migration**

Tree establishment and upward expansion of trees correlate closely with the increase in summer temperature (Almquist et al., 1998; Holtmeier and Broll, 2007; Kullman, 2007; Devi et al., 2008; 2020; Grigoriev et al., 2022). Longer growing seasons with an increase in the number of warm days lead to an increase in photosynthetic performance (Holtmeier and Broll, 2007). In addition to GSAT, winter temperature plays a critical role in influencing treeline migration. Mild winters can reduce
mortality rates among trees, thereby promoting the establishment of tree seedlings (Kulman, 2007). Conversely, late and early frosts hinder tree establishment (Holtmeier and Broll, 2007). Furthermore, the frequency of freezing cycles may pose a greater risk of frost damage to trees than the duration of exposure to low temperatures (Gross et al., 1991).

It is worth noting that the continued increase in atmospheric $CO_2$ levels is unlikely to significantly enhance tree growth or the advancement of the treeline on a global scale, as trees within the treeline ecotone are not limited by carbon availability (Körner,
2003; Wiesner et al., 2019).

Optimal soil moisture conditions are crucial for facilitating the upward migration of treelines, particularly given the substantial emergence and enhanced survival of seedlings associated with increased summer precipitation (Devi et al., 2020). Drought conditions present a substantial obstacle to seedling establishment, with limitations from drought stress likely outweighing the potential benefits of sharply rising temperatures (Bailey et al., 2021). Otherwise, both paludification and waterlogging also
impede the advancement of treelines (Crawford et al., 2003; Holtmeier and Broll, 2007). The amount of winter precipitation also has a major influence on the elevational treeline. Winters with little snow, especially in wind-exposed sites, may lead to setbacks to seedlings and saplings that are not fully protected by snow cover (Gross et al., 1991; Holtmeier, 2003; 2005a; 2005b; 2007; Kullman, 2005; Holtmeier and Broll, 2007). Snow cover during winter can offer protection to seedlings and saplings, shielding them from climatic injuries and herbivores (Holtmeier and Broll, 2007). As a result of increased winter



precipitation, forest expansion may commence earlier in snow-rich and consequently, sheltered sites (Kirdyanov et al., 2012; Hagedorn et al., 2014). Increased depth of snow cover during winter can facilitate the establishment of additional seedlings, thereby contributing to the upward expansion of trees (Germino and Smith, 2000; Smith et al., 2003; Holtmeier, 2005a; Grigoriev et al., 2022). Late-lying snow resulting from increased precipitation can significantly shorten the growing season, which may, in turn, inhibit the establishment of new generations of trees (Holtmeier and Broll, 2007). Nevertheless, delayed

snowmelt causes the growth period to begin later in the year, when temperatures are already warmer, while additional snowfall increases moisture availability during the growing season (Walker et al., 1999). In extremely snow-rich winters with increased snow load and destructive snow movements like avalanches, snow slides, and snow creep, the potential destruction of tree crowns and branches can impede tree establishment (Holtmeier and Broll, 2007; Autio, 2006; Holtmeier, 2005a). Furthermore, the presence of wet snow exacerbates damage and promotes the spread of snow fungi, potentially leading to an increase in

seedling and sapling mortality (Holtmeier, 2005a; 2005b; Holtmeier and Broll, 2007; Barbeito et al., 2013).

Wind can have very different effects on the advance of the treeline. The dispersal of seeds and pollen depends on wind speed and direction, with low to moderate wind speeds promoting the dispersal of seeds and pollen and frequent high wind speeds causing physiological and mechanical stress (Holtmeier and Broll, 2007; Kruse et al., 2018). In areas of extreme wind exposure, exacerbated by higher topography, seeds may disperse and accumulate exclusively within wind-protected shelters where seed

entry is possible (Kullman, 2005; Resler et al., 2005; Anschlag, 2006; Holtmeier and Broll, 2007; Beloiu et al., 2022). Reduced growth on the windward sides of trees, combined with frost-drying damage, underlines the damaging effects of wind exposure (Holtmeier, 1985).

As long as there are no disturbances, the height of the treeline is determined by climatic factors; recruitment-related factors are only likely to influence the local appearance and demography of the treeline (Körner, 2020). Nevertheless, the probability

of tree recruitment in the current treeline ecotone is also of high importance for the elevation of the treeline and its advance (Wiesner et al., 2019). The success of development from seed to seedling to sapling is contingent upon various factors including the availability of mature, germinable, and viable seeds, seed dispersal and dispersal method, the presence of suitable seedbeds, and the rates of establishment, growth, and survival of seedlings and saplings (Holtmeier, 1995; Juntunen and Neuvonen, 2006; Holtmeier and Broll, 2007; Holtmeier and Broll, 2019). In cold climates, short vegetation can thrive due to the relatively warm

microenvironment near the ground, while advanced tree individuals achieve tree size only if they can tolerate the surrounding air temperature (Wilson et al., 1987; Grace, 1989). Relatively warm summers promote the production of viable seeds, the establishment of seedlings and the achievement of a size that enables survival in the following winter (Karlsson and Weih, 2001; Holtmeier and Broll, 2007). In arid continental climates, particularly on fast-draining substrates, the presence of soil moisture becomes the most critical factor for the development of tree seedlings and saplings (Holtmeier and Broll, 2005; 2007).

Variability in traits enables the emergence of diverse variants and thus the adaptability to environmental changes. Through inheritance, populations adapt to their environments and promote successful traits, leading to increased population spread and resilience (Holtmeier and Broll, 2007; Gloy, 2023). The more trees in the treeline ecotone suffer from overageing (mortality caused by the effects of ageing), the more they are hindered in their ability to reproduce and at a certain age the trees die



(Holtmeier and Broll, 2007). Global and regional studies have often overemphasised broad drivers such as temperature
(Holtmeier and Broll, 2019).

The adverse effects of topography, geomorphological processes, local site conditions encompassing microclimate and soil characteristics including temperature, moisture, and mineralisation, as well as the influences of natural and anthropogenic disturbances such as mass outbreaks of leaf-eating insects or autumnal moth, and grazing activity through reindeer or other pasturing animals, may have a more pronounced impact on treeline advancement than the positive effects of increased
temperatures (Holtmeier et al., 2004; Holtmeier and Broll, 2005; 2007; Wiesner et al., 2019; Hansson et al., 2021; Brehaut et al., 2022). In many steep-sided high mountain valleys, the elevation at which treelines are found is primarily determined by the orography of the region. Factors such as high avalanche activity, mass wasting, unstable slope debris, and fragmented or absent soil cover hinder forests from reaching their maximum thermal elevational limit in these regions (Holtmeier and Broll, 2005; 2007). The treeline's advance is additionally governed by the occurrence of permafrost. Specifically, the thawing of
permafrost in elevated regions induces instability in steep mountain slopes, consequently escalating erosion processes in potential forest habitats (Lloyd, 2005; Holtmeier and Broll, 2007). Under certain conditions, wildfires can generate seedbed environments favourable for seedling emergence. However, low-severity wildfires generate diverse microsite conditions, reducing the likelihood of seeds landing on suitable seedbeds. Moreover, burned treelines may experience alterations in extreme soil temperatures and snow depths, introducing new obstacles to seedling establishment (Brehaut and Brown, 2022).

During the initial phase of treeline advancement, open tree stands are prone to experiencing higher snowpack accumulation compared to previously treeless areas (Holtmeier, 2005a).The increasing tree population density in the treeline ecotone creates a warmer microclimate, reducing the risk of summer frost damage by limiting exposure to intense solar radiation after cold nights or strong winds, thus facilitating the survival of seedlings and saplings (Örlander, 1993; Germino and Smith, 2000; Germino et al., 2002; Smith et al., 2003; Johnson et al., 2004; Holtmeier and Broll, 2007; Sigdel et al., 2020). In ecotones with
sparse tree cover during early tree invasion, snow depth and its effects may vary abruptly (Holtmeier 2005a). With increasing tree density, both intra- and inter-specific competition as well as above- and below-ground competition between trees increase. Thus, as tree density increases, the favourable effects tend to change to unfavourable effects on tree development and thus treeline migration (Kruse et al., 2016). Another significant challenge for tree migration is competition with non-arboreal vegetation, such as grassland and dwarf shrub vegetation, as they impede the access of seeds to suitable seedbeds and of
seedlings and saplings to light, nutrients and moisture (Holtmeier, 1995; Löffler et al., 2004; Dullinger et al., 2003; 2004; Holtmeier and Broll, 2007). Bare mineral soil surfaces offer favourable seed beds for wind-mediated tree seeds (Löffler et al., 2004), primarily due to the absence or reduction of competition with dwarf shrubs or grass vegetation (Hobbie and Chapin, 1998). Therefore, moderate grazing may actually provide benefits for tree establishment due to trampling, especially if exclusion of cattle or a reduction in grazing pressure follows tree seed germination (Holtmeier, 1995; Löffler et al., 2004;
Holtmeier and Broll, 2007).

The aim of the present study is to enhance our understanding of treeline dynamics and improve the predictive capabilities of vegetation models in the context of climate change. Consequently, we evaluate and contrast the relative impact of various



factors on the migration velocity of the alpine treeline. Furthermore, we integrate the impact of varying snow depths on trees in the treeline ecotone through the incorporation of a novel snow module into the LAVESI vegetation model. This integration

enables us to assess the influence of diverse snow regimes on the rate of treeline migration within the treeline ecotone.

This leads us to investigate two key research questions:

1.      What are the most significant factors influencing the migration rates of alpine treelines, and how do these factors compare in terms of their respective impacts?

2.      Does the rate of upward migration of alpine treelines within the alpine treeline ecotone vary in response to changes

170        in snow regime?

## 2 Methods

### 2.1 Model description

In our study, we used the individual-based and spatially explicit model LAVESI (*Larix* Vegetation Simulator) (Kruse et al., 2016). LAVESI, initially developed for *Larix gmelinii* in northeastern Siberia, was designed with the aim of simulating the

life cycle of larch species as completely as possible. Within one simulation step, which is one year, the relevant processes are incorporated consecutively as submodules. The previous version of LAVESI already incorporates many of the forementioned factors influencing treeline migration rates. At the beginning of each annual cycle, an 'Environment update' takes place. On the one hand, climate variables and a weather index, namely monthly means of temperature and monthly precipitation sums, are processed to estimate daily values. On the other hand, competition is assessed using an annually updated map of tree

population density. LAVESI then calculates an annual cycle of seed production, seed dispersal, establishment and competition-dependent tree growth. Finally, ageing and mortality complete a LAVESI simulation step. LAVESI has been extended to include dispersal processes related to wind speed and direction (Kruse et al., 2018), landscape topography and further boreal forest species (Kruse et al., 2022), adaptability to environmental changes by implementing trait variation and inheritance (Gloy et al., 2023), and forest fire (Glückler et al., 2024). For North America, disturbance effects caused by insects were also included

in the model.

### 2.2 Description of impact factors

In the initialisation phase of each run, the monthly weather data are processed by LAVESI (Kruse et al., 2016). The length of the growing season is calculated from the sum of days with a temperature greater than 0°C, called "net degree days" (NDD0), for each simulation year. The summer temperature is based on the sum of the temperatures of the days above 10°C, the so-

called "active air temperature" (AAT10). The average January temperature is used as an indicator for the winter temperature. Summer precipitation is the mean precipitation for the months of May to August; winter precipitation – e.g. temperature-dependent snowfall and accumulation – is part of the new snow module. A wind module, where seed and pollen dispersal are



controlled by wind speed and direction, has been implemented in LAVESI-Wind (Kruse et al., 2018). Increased tree mortality due to wind exposure is implemented in the snow module of this new version of LAVESI.

Species-dependent factors of seed production, age of maturity, number of seeds introduced, seedling establishment, and seedling mortality are already included as sub-modules in the initial model (Kruse et al., 2016). Wind-dependent seed dispersal distance, which is positively correlated with the height of the releasing tree, as well as long-distance dispersal, are part of the wind module in LAVESI-Wind (Kruse et al., 2018). The impact of overageing is considered as a specific mortality rate, which is included as a sub-module in the initial version of the model (Kruse et al., 2016). As part of the implementation of topography

and landscape sensing of individuals, the slope and the topographic wetness index (TWI) derived from a Digital Elevation Model (DEM) were first implemented in Kruse et al. (2022). The availability of the seedbed is derived from the litter layer height, the dynamics of which were first added to LAVESI in Kruse et al. (2022). The estimation of the maximum active layer thickness, which is strongly coupled to air temperature, was introduced in the original setup of LAVESI (Zhang et al., 2005, Kruse et al., 2016). Tree density dependent mortality due to competition was also included in the original version of LAVESI

(Kruse et al., 2016), whereas tree density dependent facilitation is part of the new snow module. Competition with non-arboreal vegetation such as grassland and dwarf shrub vegetation is only indirectly implemented in the seedbed availability factor.

## 2.3 New model development: The snow module

In this study, we examine the driving forces and, in particular, the constraints that influence the responses of alpine treelines. The current version of the LAVESI model successfully simulates the dynamics of the Siberian and North American latitudinal

treelines. Since many factors influencing treeline migration are common to both latitudinal and elevational treelines, we adapted LAVESI to account for the specific characteristics of alpine treelines. As part of this adaptation, various effects of snow on treeline migration are incorporated into the model to better capture the unique dynamics of alpine environments.

In recent decades, there have been a number of efforts to incorporate snow dynamics into individual-based gap models. In the SEIB-DGVM (Spatially Explicit Individual-Based DGVM) model, Sato et al. (2007) introduce a method for distinguishing

between rainfall and snowfall based on an empirical function of daily mean air temperature. This approach allows for the accumulation of snowfall in a designated snow pool, which subsequently melts in response to soil temperature variations. UVAFME (University of Virginia Forest Model Enhanced) (Wang et al., 2017) has been significantly updated from its original version FAREAST (forest gap model to simulate dynamics and patterns of eastern Eurasian forests) (Xiaodong and Shugart, 2005). However, neither individual-based gap models account for snow accumulation or melt processes. The revised version

of UVAFME by Foster et al. (2017) addresses this gap by implementing a simplified snowmelt submodel using the degree-day method proposed by Singh et al. (2000). In this framework, if daily air temperatures drop below 5 °C, precipitation is classified as snowfall and added to the snow cover; conversely, temperatures above 0 °C trigger the melting process for that day.

In previous versions of the individual-based, spatially explicit vegetation model LAVESI (Kruse et al., 2016; 2018; 2022; Gloy

et al., 2023; Glückler et al., 2024), precipitation was limited to the growing season months of May to August, neglecting any





contributions from precipitation outside this period. Consequently, processes related to snow accumulation and melt were not incorporated into the model. As part of the adaptation of LAVESI for elevational treelines, the model was enhanced to simulate variable snow scenarios. Snow depth is calculated from precipitation using a snow-to-water ratio, which can vary significantly depending on factors such as the amount of ice in the snow-producing clouds, the structure of snowflakes, and ambient

temperature. As a general guideline, we apply an average snow-to-water ratio of 10:1 (National Weather Service, 2024). Following the approach of Sato et al. (2007), processes of snow accumulation and snowmelt have also been integrated into the LAVESI model. Unlike models such as SEIB-DGVM and UVAFME, which primarily focus on temperature and moisture dynamics without fully integrating snow effects, the revised version of LAVESI explicitly considers how different snow depths influence tree growth and treeline migration.

The effects of varying snow depths on trees within the treeline ecotone have been integrated into the new snow module of LAVESI as follows.

If the height of the tree is higher than the current snow depth, the mortality rate of the tree increases by 50%. This penalty decreases linearly with increasing tree height and is eliminated once the tree reaches a height of 5 m.

When seeds are covered by snow, their germination is enhanced, exhibiting a linear increase in germination rates with greater

snow depth. Specifically, this relationship begins at zero and reaches a 30% increase in germination at a snow depth of 1 m. If seedlings and saplings are covered by snow, their growth is enhanced by 30%. The germination process is influenced linearly, either positively or negatively, based on whether the snow melts before or after the 110th day of the year. The actual day of snowmelt is divided by 110, and the deviation from 1.0 is added to the germination probability.

Increased snow load and avalanche occurrences during a given year contribute to higher mortality rates and reduced seed

production in the following years. The probability of damage due to extreme snow load is calculated based on stochastic daily snowfall events that exceed the 99th percentile for a given location. When this threshold is surpassed, affected trees experience a 50% increase in mortality. Avalanche risk is determined by the maximum snow depth during a winter season. The top 0.1% of total snow depths occurring over a historical time series in the simulation run forms a threshold, which is compared with the current winter season's maximum snow depth to determine the avalanche risk. The risk increases linearly as snow depth

rises above the threshold. Mortality is added where a modelled avalanche impacts the leading edge of tree stands, with its destructive force diminishing downslope due to protecting tree growth. Trees that are affected by an avalanche will produce fewer seeds for 10 years, depending on the avalanche's force and the corresponding damage to the tree.

The influence of wind exposure on tree mortality has been included in the snow module. The wind exposure of a tree depends on its elevation and the density of surrounding trees. If trees are highly exposed without surrounding groups of trees to protect

them from the wind, their mortality will increase and vice versa.

The negative effect of increasing tree density on tree survival was already included in the model using competitive strength evaluations (Kruse et al., 2016). Tree mortality depends on tree density due to competition, i.e. as tree density increases, competition increases and tree mortality increases. However, the positive effect of tree density on tree survival, e.g. by providing a protective wind shelter or promoting snow accumulation, is first included in this study. The facilitation strength is





influenced by tree density, i.e. as tree density increases, facilitation increases. With increasing facilitation, tree density-dependent mortality is reduced linearly, reaching its maximum reduction at a tree density of 0.5 (dimensionless). With further increasing tree density, facilitation strength decreases again until ceases entirely at a tree density of 1.0. Beyond this threshold, when the tree density exceeds 1.0, only competition occurs.

The increased moisture availability during the growing season, resulting from enhanced winter snowfall, is incorporated into 265 the general weather processing during the initialisation phase. This adjustment modifies the species-specific weather index, thereby influencing the climate-growth response of individual tree species.

The LAVESI-Mountain Treelines 1.0 model code, along with the input data used in this study, is available for review on GitHub (https://github.com/StefanKruse/LAVESI/tree/circumboreal_mountain). Upon manuscript acceptance for publication, the code, input data, and simulation output will be permanently archived on Zenodo.

**2.4 Study sites**

**Table 1: Characteristics of the study sites.**

|  | Site CA – Fort McPherson West | Site RU – Lake Ilirney | Site AK – Road to Central |
|---|---|---|---|
| Country | Canada | Russia | Alaska/ USA |
| Location (Coordinates) | xmin = -136.001; xmax = -135.867; ymin = 67.135; ymax = 67.187 | xmin = 168.128; xmax = 168.501; ymin = 67.301; ymax = 67.437 | xmin = -145.552; xmax = -145.440; ymin = 65.420; ymax = 65.467 |
| Mean annual temperature (°C) (1990-2020) | -5.36 | -8.61 | -0.841 |
| Total annual precipitation (mm) (1990-2020) | 221 | 222 | 178 |
| Vegetation type | White spruce single trees on steep slope with avalanche damage | Tundra–taiga ecotone, characterised by diverse vegetation: open tundra, shrublands and forest tundra | West to North upslope, a little bit wet, White and Black spruce mixed forest and many *Salix* and *Betula* shrubs |
| Tree species in the region | *Larix laricina* *Picea glauca* *Picea mariana* *Betula papyrifera* *Populus tremuloides* *Populus balsamifera* *Pinus contorta* | *Larix gmelinii* *Larix sibirica* *Larix cajanderi* *Picea obovata* *Pinus sylvestris* *Pinus sibirica* *Betula pendula* | *Larix laricina* *Betula pendula* subsp. *mandshurica* *Tsuga mertensiana* *Picea mariana* *Picea sitchensis* *Picea glauca* *Populus tremuloides* *Populus balsamifera* |



**Table 2: Adjustments to LAVESI simulation settings across study sites.**

| Plot code / weatherchoice (parameters.txt) | roi (parameters.txt) | seedintroarea (parameters.txt) | simulation area (inc/declaration.h) |
|---|---|---|---|
| 5004 - Fort McPherson West | 2 – CA | seedintro_miny=600; seedintro_maxy=700; seedintro_minx=0; seedintro_maxx=50; | constexpr unsigned int treerows = 990; constexpr unsigned int treecols = 105; |
| 5005 - Lake Ilirney | 1 – RU | seedintro_miny=600; seedintro_maxy=700; seedintro_minx=0; seedintro_maxx=50; | constexpr unsigned int treerows = 1005; constexpr unsigned int treecols = 105; |
| 5009 - Road to Central | 3 – AK | seedintro_miny=900; seedintro_maxy=1000; seedintro_minx=25; seedintro_maxx=75; | constexpr unsigned int treerows = 1005; constexpr unsigned int treecols = 105; |


The model had to be parameterised for the three study sites in Fort McPherson West, Canada (CA), at Lake Ilirney, Russia (RU), and at the Road to Central, Alaska, USA (AK) (see Table 1), for which the respective adjustments were made in the input files parameter.txt and declaration.h (see Table 2). Satellite images of the study sites, along with a digital elevation model (DEM), slope, and topographic wetness index (TWI) for the simulation area, are provided in Figures A1–C1 in the Appendix.

**2.5 Model inputs and simulation scenarios**

Monthly precipitation and temperature data for the historical period (1400–1950 CE) were obtained from the Max Planck Institute Earth System Model (Transient MPI-ESM Glac1D_P3) (Kapsch et al., 2022). This past climate data was bias-corrected to align with modern data from the CRU-TS dataset (Harris et al., 2020; available for 1900–2020 CE), ensuring consistent means during the overlap period (1900–1950 CE) to avoid discontinuities. The corrected dataset was extended to
2020 CE using CRU-TS. The first 620 years are both to simulate the current conditions of the populations and as the spin up-phase to allow the forests to establish from seeds introduced over the entire area. Climate projections for 2020–2100 CE were derived from the CMIP6 MPI-ESM-2-LR model (Eyring et al., 2016; specific version: r1i1p1f1) and similarly bias-corrected to CRU-TS, using the overlap period (2015–2020 CE) to ensure consistency.

Monthly wind data for 1400–1950 CE was obtained from the transient MPI-ESM Glac1D_P3 model. Since the vegetation
model LAVESI required 6-hourly wind data, a downscaling approach was applied. Modern wind data from the ERA5 dataset



(Hersbach et al., 2023), covering 1940–2020 CE, was used to create a 6-hourly resolution dataset for the overlapping period and extend the dataset to 2020 CE. ERA5's detailed temporal resolution facilitated realistic downscaling and interpolation. Future wind data from the CMIP6 MPI-ESM-2-LR model (2020–2100 CE), provided at daily resolution, was interpolated to 6-hourly means to meet LAVESI's requirements.

The environmental input data for LAVESI was derived from the Ensemble Digital Terrain Model (EDTM) with a 30 m resolution (Ho et al., 2023). Key information, including elevation, slope, and the topographic wetness index (TWI), was extracted from the DEM of the simulation area.

Simulation runs are performed using a transect representing a slice of the alpine treeline with a size of 10 km x 0.1 km.

To evaluate and compare the influence of various factors on the migration rate of the alpine treeline and to identify key drivers

and assess how variations in these factors influence treeline dynamics, a sensitivity analysis was performed. Most of the factors examined in this study have already been implemented in the original LAVESI model (Kruse et al., 2016) or its subsequent versions. These factors are detailed extensively in the corresponding publications. The only newly introduced component in this study was the snow module, which is specifically designed to simulate the impact of snow cover dynamics on the treeline. A description of this snow module is provided earlier in this manuscript. For the sensitivity analysis following Kruse et al.

(2018), simulations were run in which the following 17 factors were increased by 10% and decreased by 10% compared to an average reference value: growing season length, summer temperature, winter temperature, summer precipitation, wind speed, seed production, maturation age, seed intro number permanent, dispersal distance, long distance dispersal, seedling establishment, seedling mortality, overageing mortality, slope, twi, seed bed availability, and active layer.

In addition, further simulations were run with the forest fire module (Glückler et al., 2024), the insect disturbance module, or

the module for trait variation and inheritance (Gloy et al., 2023). The results of each of the simulations were compared with the results of the reference run.

Finally, simulations were conducted using the new snow module, with a 10% increase and decrease applied to the following factors within the module: winter precipitation, wind exposure, and facilitation. The results were then compared to those obtained from the reference snow module run.

We compared the effect of all these settings on treeline migration rate, where the treeline is defined as the northernmost position where forest density does not fall below a threshold of one hundred trees per hectare, where a tree must be >1.3 m tall.

In total, 45 different simulation scenarios were computed (for a structured overview, see Table D1 in the Appendix). These include those evaluating model sensitivity to impact factors, as described before, as well as the simulations with and without the forest fire module, the insect disturbance module, the module for trait variation and inheritance, and the new snow module

including the evaluation of the model sensitivity to impact factors within the snow module.

Each of the simulation scenarios was repeated 40 times.





**2.6 Statistical analyses of simulation**

The output files generated by the LAVESI model provide information on stem count and tree density for each 5×5 m grid cell within the simulation area. The statistical analysis was performed using the statistical tool R (R Core Team, 2023).

Stem counts for all tree species were aggregated across spatial positions for every fifth year between 2020 and 2095. The aggregated data are visualised as line graphs using the 'ggplot2' R package (Wickham, 2016) to illustrate changes over time across different scenarios. Additionally, spatial distributions of tree densities are depicted with separate raster plots created for the years 2020 CE, 2055 CE, and 2095 CE. Treelines were determined using threshold-based criteria of one hundred trees per hectare.

The migration rate of the alpine treeline was recorded for each simulation by calculating the change in treeline position between 2020 and 2095 for each scenario. Using the *diff* function in R, we measured incremental changes between consecutive treeline positions, capturing the position change between each measured interval. These differences were then averaged to obtain the mean migration rate, which represents the average advancement of the treeline over time. The significance of the migration rates was assessed using a paired t-test, conducted with the *t.test* function in R.

The relative change in migration rate was calculated for both the increased and decreased scenarios to quantify the sensitivity of the treeline to each factor. All resulting migration rates were computed to sensitivity values that present the percentage of change in migration rate scaled to the 10% change in the factor. The sensitivity values are calculated from the following Eq. (1):

$$S = \frac{\Delta Y \times 100}{\Delta P},$$  (1)

where $\Delta Y$ represents the change in the migration rate derived from each simulation run and $\Delta P$ is the 10% change in the parameter. The significance of the sensitivity values was evaluated using a one-sample t-test, performed with the *t.test* function in R (R Core Team, 2023).

We conducted a redundancy analysis (RDA) using the *rda* function from the 'vegan' R package (Oksanen et al., 2018) to assess the relationships between environmental variables and treeline migration rates across the three study sites (CA, RU, and AK). The response variable consisted of treeline migration rates from 45 different simulation scenarios. In these scenarios, key factors influencing treeline migration were manipulated by switching them on or off, or by increasing or decreasing their magnitude compared to an average reference value. Each scenario was simulated across 40 runs per study site. Environmental

predictor variables were derived from weather data for each study site, spanning the period from 1990 to 2020. The climate variables, including mean annual temperature, mean January temperature, mean July temperature, mean summer (JJA) temperature, accumulated annual temperature (AAT), degree days above threshold (DDT), total annual precipitation, and summer (JJA) precipitation, were calculated and averaged for each study site. Stepwise forward selection, using the *ordistep* function from the 'vegan' R package, was performed to identify the most significant climate predictors of treeline migration

 

rates. The process began with a null model and progressively added variables based on the Akaike Information Criterion (AIC) and F-statistics. Significant variables (p < 0.05) were retained in the final model, with significance evaluated using 999 permutations. For the RDA plot, site (migration rates), species (scenario), and environmental (weather data) scores were extracted and visualised using the 'ggplot2' R package.

## 3 Results

### 3.1 Comparative analysis of factors influencing alpine treeline migration

**Table 3: Sensitivity analysis results for treeline migration rates. Mean sensitivity values (in %/%) and standard deviations for 17 impact factors, each adjusted by ±10% from an average reference value, resulting in 34 distinct scenarios across three different study sites. * Highly significant with p < 0.05, and the remaining values non-significantly different from the reference run. Values are the mean of 40 simulations.**

| | Fort McPherson West (Canada) | | Lake Ilirney (Russia) | | Road to Central (Alaska) | |
| --- | --- | --- | --- | --- | --- | --- |
| | 10% | | 10% | | 10% | |
| Impact factor | S- | S+ | S- | S+ | S- | S+ |
| Activelayer | -3.65±9.41* | 2.54±10.01 | -1.23±6.64 | 0.98±6.71 | -8.29±40.82 | 6.64±29.44 |
| Dispersaldistance | -0.75±8.02 | 2.99±9.85 | -0.37±5.24 | 0.60±5.48 | -6.70±28.06 | 6.36±39.29 |
| Growingseasonlength | -3.12±9.66* | 2.78±8.30* | -2.24±7.02 | 1.59±6.79 | -3.55±14.66 | -0.40±10.06 |
| Longdistancedispersal | -1.87±8.85 | 1.73±7.88 | -0.44±6.22 | 2.90±6.08* | -9.98±51.50 | 2.91±15.23 |
| Maturationage | -3.84±8.03* | 0.30±7.18 | -3.10±7.39* | 1.91±7.81 | -5.03±40.20 | 7.93±40.25 |
| Overageingmortality | -2.69±10.35 | 3.22±8.86* | 0.12±4.32 | 1.97±7.72 | -10.70±65.20 | 6.40±39.20 |
| Seedbedavailability | -2.06±8.58 | 4.32±9.66* | 0.50±6.08 | 2.51±7.71* | -5.95±35.82 | 4.60±29.80 |
| Seedintronumberpermanent | -0.77±6.45 | 2.12±8.57 | -0.95±6.32 | 1.65±7.03 | 0.17±7.64 | 5.14±27.14 |
| Seedlingestablishment | -1.01±7.86 | 3.39±9.39* | -1.31±8.65 | 2.83±7.84* | -7.38±46.34 | 5.68±36.30 |
| Seedlingmortality | -2.34±9.50 | 2.35±7.15* | -2.11±6.13* | 1.45±7.22 | -9.40±46.04 | 4.89±25.42 |
| Seedproduction | -1.18±7.63 | 4.50±8.84* | -1.33±8.13 | 1.52±6.57 | -8.59±37.65 | 1.51±15.73 |
| Slope | -1.96±7.90 | 0.24±8.06 | -2.03±7.01 | 1.25±6.60 | -7.70±57.74 | 8.37±44.68 |
| Summerprecipitation | -3.03±8.99* | 0.43±6.28 | -1.19±5.16 | 2.25±7.21 | -6.08±25.64 | 3.94±14.03 |
| Summertemperature | -2.38±8.85 | 2.73±9.19 | -0.67±5.73 | 1.24±8.19 | -3.54±18.97 | 4.33±32.80 |
| Twi | -2.74±9.71 | 4.00±10.29* | -1.22±7.09 | 1.71±8.66 | -4.51±29.31 | 3.28±16.66 |
| Windspeed | -2.52±9.80 | 2.99±8.99* | -1.31±6.41 | 2.61±7.53 | -10.05±62.16 | 4.09±22.37 |
| Wintertemperature | -2.18±7.38 | 2.61±7.35* | -0.20±4.67 | 2.10±7.38 | -9.32±43.66 | 12.38±76.15 |
| Mean | -2.30 ± 8.43 | 2.98 ± 8.76 | -0.97 ± 6.39 | 1.89 ± 7.14 | -7.21 ± 39.51 | 5.22 ± 32.45 |


In general, the sensitivity of treeline migration rates to ±10% changes in impact factors was assessed across all three study sites. On average, a 10% increase in impact factors results in a +2.98% change in migration rate (SD = 8.76) at site CA, +1.89% (SD = 7.14) at site RU, and +5.22% (SD = 32.45) at site AK. A 10% decrease leads to changes of -2.30% (SD = 8.43) at site




CA, -0.97% (SD = 6.39) at site RU, and -7.21% (SD = 39.51) at site AK. The minimum and maximum significant sensitivities observed are +0.43% and +4.50% for site CA and +0.50% and +2.90% for site RU. For site AK, no significant values were detected (Table 3).

Site AK exhibits the highest mean sensitivity values; however, these results are characterised by substantial variability, preventing any significant findings. In contrast, site RU generally shows lower sensitivity magnitudes across all scenarios, although some scenarios do reach significance. Site CA displays several scenarios with significant positive or negative sensitivities, indicating a stronger influence on migration dynamics compared to the other sites.

The direction of sensitivity across all study sites is consistent for the majority of impact factors, with positive and negative sensitivities aligning similarly between sites. There are four exceptions where the direction of influence differs between study sites, but these deviations are minimal and not statistically significant.

Notably, four scenarios demonstrate significant sensitivity at both sites CA and RU ($p < 0.05$). Increased seedbed availability has the most pronounced positive influence on migration, with mean sensitivities of 4.32% at site CA and 2.51% at site RU (seedbedavailability_plus10%). In contrast, reduced maturation age strongly hinders migration potential, as reflected by negative mean sensitivities of -3.84% at site CA and -3.10% at site RU (maturationage_minus10%). Favourable conditions for seedling establishment also plays an important role, promoting migration with mean sensitivities of 3.39% at site CA and 2.83% at site RU (seedlingestablishment_plus10%). Lastly, higher wind speeds facilitate migration through enhanced seed dispersal, as indicated by mean sensitivities of 2.99% at site CA and 2.61% at site RU (windspeed_plus10%).





## 3.2 Impact of snow-related processes on alpine treeline migration rates

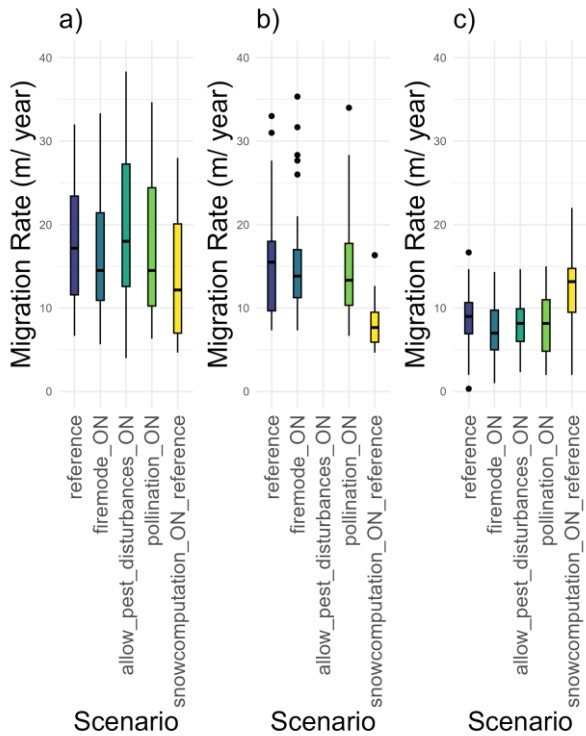

**Figure 1: Box plots showing migration rates from 40 simulation runs at three study sites: a) Fort McPherson, Canada (CA), b) Lake Ilirney, Russia (RU), and c) Road to Central, Alaska, USA (AK). Each panel compares the reference scenario with four scenarios in which specific environmental modules were activated. Note that for the Russian study site, the insect disturbance module was not implemented in LAVESI.**

The results of additional simulations, which were performed with the forest fire module (Glückler et al., 2024), the insect disturbance module, the trait variation and inheritance module (Gloy et al., 2023), and the snow module implemented in this study, were compared with the results of the reference run (Figure 1). Notably, across all three study sites, only the simulations incorporating the influence of snow-related factors resulted in a significant change in the treeline migration rate.



**Table 4: Sensitivity analysis results for treeline migration rates within snow computation simulations. Mean sensitivity values (in %/%) and standard deviations for three impact factors, each adjusted by ±10% from an average reference value, resulting in six distinct snow computation scenarios across three different study sites. * Highly significant with p < 0.05, and the remaining values non-significantly different from the reference run. Values are the mean of 40 simulations.**

| | Fort McPherson West (Canada) | | Lake Ilirney (Russia) | | Road to Central (Alaska) | |
|---|---|---|---|---|---|---|
| | 10% | | 10% | | 10% | |
| Impact factor | S- | S+ | S- | S+ | S- | S+ |
| facilitation | -5.26±12.64* | 6.53±14.38* | -2.03±6.79 | 4.15±8.68* | -4.48±15.69 | 2.37±13.34 |
| windexposure | -5.19±13.42* | 2.62±10.53 | -6.07±9.05* | -1.23±2.85* | -1.54±6.60 | 4.79±14.34* |
| winterprecipitation | -7.64±14.99* | 3.91±11.41* | -2.43±7.73 | 2.99±9.09* | -1.96±14.64 | 5.30±12.15* |
| Mean | -6.49 ± 14.47 | 4.35 ± 12.44 | -3.51 ± 7.86 | 1.30 ± 6.87 | -2.66 ± 12.31 | 4.15 ± 13.28 |


The sensitivity of treeline migration rates to ±10% changes in snow-related impact factors, relative to the snow module reference run, was analysed across all three study sites. On average, a 10% increase in impact factors leads to a +4.35% change in migration rate (SD = 12.44) at site CA, +1.30% (SD = 6.87) at site RU, and +4.15% (SD = 13.28) at site AK. Conversely, a 10% decrease in impact factors results in a -6.49% change (SD = 14.47) at site CA, -3.51% (SD = 7.86) at site RU, and -

2.66% (SD = 12.31) at site AK. The minimum and maximum significant sensitivities observed are +3.91% and +6.53% for site CA, +2.99% and +4.15% for site RU, and +4.79% and +5.30% for site AK (Table 4).

At site CA, several scenarios demonstrate strong positive or negative sensitivities, with nearly all results reaching statistical significance (p < 0.05). In contrast, site RU exhibits more moderate sensitivities, although some scenarios still achieve statistical significance. Despite relatively high mean sensitivity values at site AK, substantial variability in the results causes

fewer statistically significant outcomes

The sensitivity analysis for snow computation reveals consistent directional influences across the three study sites, with one notable exception: windexposure_plus10%. While sites CA and AK exhibit positive sensitivity values, site RU displays a negative sensitivity value. The sensitivity values at sites RU and AK are statistically significant (p < 0.05).

The scenario winterprecipitation_plus10% has statistically significant impacts across all sites, suggesting that increased snow

accumulation during winter enhances migration potential. Both facilitation_plus10% and windexposure_minus10% show significant sensitivity at sites CA and RU (p < 0.05). Increased facilitation positively influences migration potential, with the strongest effects observed at site CA, while reduced wind exposure negatively impacts migration at both sites. Although site AK displays similar patterns, the variability in the data prevented statistical significance.



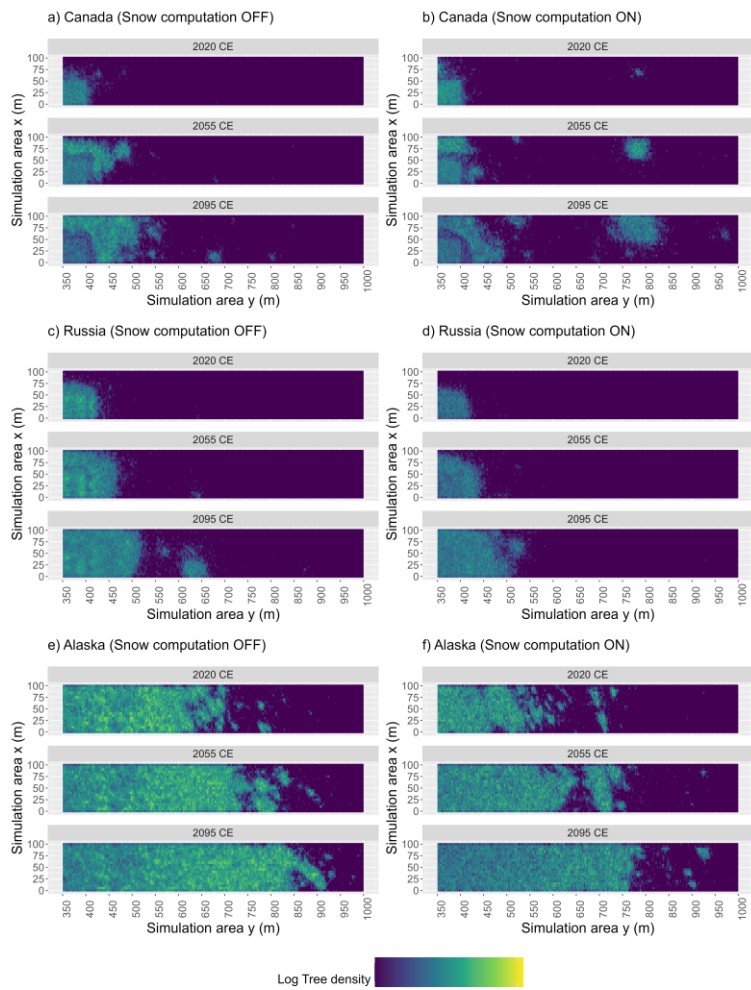

**Figure 2: Tree density over time. Tree density in 2020, 2055, and 2095 CE at study site Fort McPherson West, Canada (CA) under a) the reference scenario (run 34) and b) the scenario with activated snow computation (run 2). Tree density in 2020, 2055, and 2095 CE at study site Lake Ilirney, Russia (RU) under c) the reference scenario (run 13) and d) the scenario with activated snow computation (run 2). Tree density in 2020, 2055, and 2095 CE at study site Road to Central, Alaska, USA (AK) under e) the reference scenario (run 15) and f) the scenario with activated snow computation (run 3). (Run with the median migration rate calculated from 40 runs for each simulation scenario). Each plot represents data from the simulation run with the median migration rate derived from 40 simulation runs per scenario. Tree density values were log-transformed to enhance data visualisation.**

Across all three study sites, simulations without the snow module (Snow computation OFF) produce different tree density values over time compared to those incorporating snow-related processes (Snow computation ON). For all sites, the inclusion of snow processes results in an increase in tree density by the years 2055 and 2095 (Figure 2).





**3.3 Relationships between environmental variables and treeline migration rates**

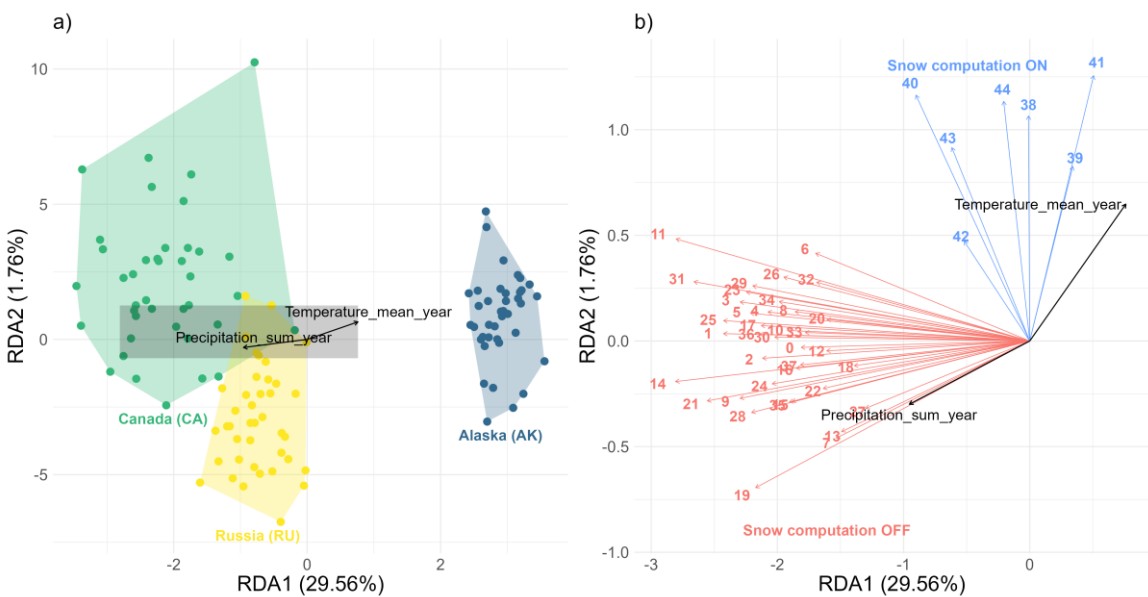

**Figure 3: Redundancy Analysis (RDA) biplot showing a) site scores and b) species scores along the first two principal axes. Plot b provides a close-up of the centre region (grey rectangle) of plot a to highlight tightly grouped species. Site scores (migration rates) are represented by points, colour-coded according to their study site. Species scores (simulation scenarios) are depicted as arrows, colour-coded to indicate whether snow computation is activated (blue: Snow computation ON) or not (red: Snow computation OFF). Black arrows in both plots represent the direction and strength of environmental variables influencing migration rates.**

The redundancy analysis (RDA) identified two significant constrained axes, collectively explaining 31.32% of the variance in treeline migration rates across simulation scenarios and study sites (Figure 3). The site scores reveal distinct clusters of study sites along the RDA1 axis, reflecting geographical differences. Sites CA and RU group closely together on one side of the axis, while site AK is distinctly separate on the opposite side (Figure 3a). Along the RDA2 axis, the species scores indicate that scenarios incorporating snow-related processes (Snow computation ON) are clearly distinguished from those without snow effects (Snow computation OFF) (Figure 3b).

The stepwise forward selection in RDA identified total annual precipitation and mean annual temperature as the most significant climate predictors of treeline migration rates across the simulation scenarios and study sites. Total annual precipitation is the first variable included in the model (AIC = 903.76, F = 43.80, p = 0.001), followed by mean annual temperature (AIC = 898.55, F = 7.25, p = 0.001). Total annual precipitation shows a positive correlation with treeline migration rates at sites CA and RU, but a negative correlation at site AK. Conversely, mean annual temperature exhibits an opposite effect: positively correlating with migration rates at site AK and negatively correlating at sites CA and RU.



## 4 Discussion

**Table 5: Expected and observed direction of influence of impact factors on migration rate.**

| Impact factor | Expected influence | Observed influence | Verification status |
|---|---|---|---|
| Active layer depth (activelayer) | Positive | Positive | Confirmed |
| Seed and pollen dispersal (dispersaldistance) | Positive | Positive | Confirmed |
| Length of the growing season (growingseasonlength) | Positive | Positive | Confirmed |
| Seed and pollen dispersal over long distances (longdistancedispersal) | Positive | Positive | Confirmed |
| Age at which a tree becomes mature (maturationage) | Negative | Positive | Not confirmed |
| Mortality of trees caused by the effects of ageing (overageingmortality) | Negative | Positive | Not confirmed |
| Availability of suitable seedbeds (seedbedavailability) | Positive | Positive | Confirmed |
| Permanent seed introduction (seedintronumberpermanent) | Positive | Positive | Confirmed |
| Establishment of seedlings (seedlingestablishment) | Positive | Positive | Confirmed |
| Mortality of seedlings (seedlingmortality) | Negative | Positive | Not confirmed |
| Factor of seed productivity (seedproduction) | Positive | Positive | Confirmed |
| Slope angle (slope) | Negative | Positive | Not confirmed |
| Summer precipitation (summerprecipitation) | Variable, magnitude-dependent | Positive | Partly confirmed |
| Summer temperature (summertemperature) | Positive | Positive | Confirmed |
| Topographic wetness index (twi) | Positive | Positive | Confirmed |
| Wind speed (windspeed) | Variable, magnitude-dependent | Positive | Partly confirmed |
| Winter temperature (wintertemperature) | Positive | Positive | Confirmed |
| Facilitation on tree growth through snow (Snowcomputation_ON_facilitation) | Positive | Positive | Confirmed |
| Wind exposure (Snowcomputation_ON_windexposure) | Variable, magnitude-dependent | Variable, magnitude-dependent | Confirmed |
| Temperature-dependent snow fall and accumulation (Snowcomputation_ON_winterprecipitation) | Variable, magnitude-dependent | Positive | Partly confirmed |

### 4.1 Site-specific responses to factors influencing alpine treeline migration

#### 4.1.1 Spatial variability in migration rates

The RDA analysis reveals a clear spatial grouping of locations based on migration rates, with sites CA and RU clustering together, while site AK is distinctly separate. This pattern underscores the influence of unique environmental conditions at
each site on migration dynamics. Such geographical and ecological variability in treeline responses aligns with previous findings, reinforcing site-specific factors as the central factor to shaping treeline migration (Harsch et al., 2009; Hansson et al., 2021; 2023).





### 4.1.2 Sensitivity analysis: site-specific patterns

The sensitivity analysis further uncovered marked differences in responses to ±10% changes in impact factors across the three sites (CA, RU, AK). Site AK exhibited high mean sensitivity values but with substantial variability, reflecting the influence of complex and less predictable interactions between ecological and climatic factors. This variability makes migration dynamics at site AK inherently more uncertain compared to the more consistent and significant sensitivities observed at sites CA and RU. These findings underscore the need to account for localised environmental conditions when predicting treeline responses to climate change.

### 4.1.3 Consistency and anomalies in sensitivities

Despite these site-specific differences, the overall consistency in the direction of sensitivities across all sites enhances confidence in the key drivers of treeline migration identified by this study.

While many observed sensitivities align with ecological expectations, some findings are unexpected and warrant further investigation to clarify these interactions (see Table 5). For instance, the positive relationship between maturation age and

migration rate is counterintuitive, as earlier maturation typically shortens generation time and reduces exposure to juvenile mortality before the first reproductive event (Stearns, 2000). However, later maturation offers advantages such as larger tree size at maturity and higher fecundity (Stearns, 2000). Moreover, taller, older trees likely enhance migration potential through improved pollen and seed dispersal distances, consistent with the dispersal model used (Kruse et al., 2018). This extended dispersal likely reduces local competition among seedlings, enhancing migration potential, even when maturation age

increases.

Another unexpected finding is the association of higher seedling and overageing mortality with increased migration rates. One explanation could be that, while higher mortality intuitively reduces population growth, it may simultaneously reduce competition and increase resource availability for surviving seedlings, thereby enhancing their establishment and growth (Kruse et al., 2016). This highlights the complex balance between population dynamics and resource competition in

determining migration outcomes.

The positive sensitivity of migration rates to slope angles also stands out, as steeper slopes are typically considered physical barriers to migration. This may be attributed to the creation of diverse microhabitats in complex terrain, which reduce competition and provide unique niches for tree establishment, as observed by Soliveres et al. (2011).

### 4.1.4 Important factors for treeline migration

The significant sensitivity observed at sites CA and RU underscores the importance of certain environmental factors in promoting treeline migration. For example, reducing maturation age negatively impacts migration potential, likely because of the reduced tree height at maturity, which limits pollen and seed dispersal distances as discussed earlier (Stearns, 2000; Kruse et al. 2018).



Seedbed availability emerges as a crucial factor for migration potential, with sensitivity analysis confirming that increases in suitable seedbeds significantly enhance the migration rate, as expected. Suitable seedbeds provide favourable conditions for tree establishment, especially bare mineral soils, with their reduced competition from dwarf shrubs or grasses, which offer ideal conditions for wind-mediated tree seeds (Löffler et al., 2004; Hobbie and Chapin, 1998; Holtmeier and Broll, 2007).

Seedling establishment strongly supports treeline advancement, as higher rates of successful seedling establishment drive treeline progression and maintain population stability (Holtmeier and Broll, 2007), though delays in processes such as seed dispersal, establishment, or growth may slow migration rates, as noted by Holtmeier and Broll (2019).

Wind speed is another important factor influencing treeline migration, as moderate wind velocities enhance seed and pollen dispersal, aligning with findings from Holtmeier and Broll (2007) and Kruse et al. (2018).

These findings align closely with existing literature, highlighting the complex interplay between factors such as maturation age, seedbed availability, seedling establishment, and wind speed in shaping treeline migration processes. The observed patterns are consistent with broader ecological principles related to tree migration under changing environmental conditions. However, the site-specific variations in sensitivity emphasise the critical need to account for localised conditions in predictive simulation models.

Overall, the simulations analysing various factors affecting alpine treeline migration reveal statistically significant influences on migration rates at sites CA and RU, offering valuable insights into the key factors driving migration potential under current conditions. In contrast, the variability observed at site AK underscores the need for further investigation into local factors influencing migration sensitivity. These results collectively reinforce the importance of integrating both broad-scale ecological drivers and site-specific conditions to enhance our understanding of treeline dynamics.

## 4.2 Climate drivers of alpine treeline migration across study sites

Our findings align with previous research showing that climate-induced treeline shifts are shaped by a complex interplay of warming temperatures and precipitation variability, which jointly influence seedling establishment and growth (Kullman, 2005, 2007; Holtmeier and Broll, 2005, 2007; Körner, 2012). Here, total annual precipitation and mean annual temperature emerge as the most influential drivers of treeline migration accounting for over 30% of the observed variance in migration rates across the different scenarios. Traditionally, growing season temperature has been regarded as a critical factor driving treeline shifts and forest expansion (Dial et al., 2007; Devi et al., 2008; 2020). However, our findings underscore the importance of considering year-round climate conditions in shaping treeline dynamics, a perspective consistent with studies emphasising the importance of mean annual temperature in influencing treeline migration (Cazzolla Gatti et al., 2019; Danby and Hik, 2007).

In the Polar Urals, winter precipitation has more than doubled over the 20[th] century (Devi et al., 2020), while in the Kenai Mountains, Alaska, mean annual temperatures have risen by 2°C since the 1950s, with winter temperatures increasing by as much as 4°C (Dial et al., 2007). These trends underscore the necessity of considering winter climate, in addition to growing season conditions, when studying treeline dynamics.





Although precipitation outside the growing season is often simply added to its annual sum, primarily in relation to soil water balance, our results highlight the multifaceted role of snow, which extends beyond its contribution to water availability (see section 4.3 The role of snow in alpine treeline dynamics).

### 530 4.2.1 Site-specific responses to climate drivers

The migration responses to the climate drivers temperature and precipitation vary between the sites. Site AK is characterised by relatively higher mean annual temperatures and lower precipitation compared to sites CA and RU.

At site AK, warmer mean annual temperatures positively influence migration. This effect is likely due to an extended growing season and improved soil warmth, which enhance seedling germination and establishment. These findings align with evidence 535 suggesting that warmer conditions promote tree germination, establishment, and survival during the growing season (Almquist et al., 1998; Holtmeier and Broll, 2007; Kullman, 2007; Devi et al., 2008; 2020; Grigoriev et al., 2022). In contrast, at the cooler, wetter sites CA and RU, higher temperatures are associated with reduced migration rates. This negative effect may be attributed to drought stress, that may override temperature benefits at upper treeline boundaries, as warming-induced soil moisture limitations could hinder tree growth and prevent treeline advance in a warmer world (Bailey et al., 2021; Gruber et 540 al., 2022).

The influence of precipitation also varies across the sites. At sites CA and RU, higher precipitation is positively correlated with accelerated migration rates. This is likely due to improved soil moisture, which supports seedling establishment and early growth in these regions (Holtmeier and Broll, 2007). These findings are consistent with studies showing that treeline migration and forest expansion are linked to increased moisture availability, particularly from higher winter precipitation (Hagedorn et 545 al., 2014; Grigoriev et al., 2022). Conversely, at site AK, higher precipitation is associated with slower migration rates, possibly due to negative factors such as waterlogging or paludification implicitly modelled by an optimum precipitation and adverse effects on potential growth in the model (Kruse et al., 2016). These findings highlight the potential for moisture-related limitations on tree expansion under certain site conditions (Crawford et al., 2003; Holtmeier and Broll, 2007).

Additionally, the different tree species present across sites likely play a role in shaping these site-specific responses. Species- 550 specific stand structure dynamics influence how various species respond to environmental conditions, further contributing to the observed variation in migration rates (Chhetri et al., 2020).

### 4.3 The role of snow in alpine treeline dynamics

### 4.3.1 Significant impact of snow-related processes on alpine treeline migration

Our study further demonstrates that snow-related processes, incorporated into the LAVESI snow module, consistently reveal 555 significant changes in migration rates across all three study sites. This finding aligns with the work of Huang et al. (2023), who show that seasonal snow-cover patterns – particularly spring snowline elevation and snow-cover duration – affect treeline elevation in the eastern Himalaya. Similarly, Niittynen and Luoto (2018) demonstrate that incorporating snow persistence data





into species distribution models enhances predictive accuracy and refines distribution mapping in high-elevation and high-latitude ecosystems. Additionally, Autio (2006) predicts that warmer temperatures could lead to increased snowfall, potentially resulting in heavier snow loads that may damage or even destroy tree crowns and branches. Collectively, these findings underscore the critical role of snow as an environmental predictor shaping treeline dynamics in alpine and subarctic ecosystems.

In contrast, recent studies evaluating environmental modules with the model LAVESI, such as explicit forest fire (Glückler et al., 2024), biotic (insect) disturbances, and trait variation and inheritance (Gloy et al., 2023), have highlighted the significance of various processes in influencing tree migration. However, our simulations do not reveal notable impacts from these factors at alpine treeline ecotones. This discrepancy may be attributed to the relatively short duration of our simulations, which might not have allowed enough time for these processes to fully manifest. For the study sites of our study, findings emphasise the immediate and dominant influence of snow-related processes on treeline dynamics, compared to longer-term processes like fire or trait evolution.

### 4.3.2 The dual role of snow and model enhancements

The separation between scenarios with and without snow effects along the second RDA axis further underscores the substantial impact of snow-related processes on migration rates. Snow acts as both a facilitator and a hindrance, depending on environmental context. Incorporating snow effects is thus critical for accurately assessing treeline migration in snow-dominated regions (e.g. Niittynen and Luoto, 2018; Huang et al., 2023).

Compared to earlier implementations of snow effects in vegetation models, our new snow module represents a significant advancement. Instead of merely adding precipitation outside the growing season to the annual total, the module captures the complex and multifaceted role of snow, extending beyond its contribution to water availability. Hansson et al. (2023) note that average precipitation changes often show weak correlations with treeline migration, likely due to model limitations that neglect variables like moisture stress and extreme precipitation events. Similarly, Kharuk et al. (2022) observe that under harsh conditions, factors such as wind and soil moisture often outweigh the role of snow. By incorporating previously overlooked factors such as wind exposure and microtopographical features, the enhanced module provides a more nuanced representation of treeline dynamics, capturing the complex interactions between snow and other environmental drivers. These processes – modelled on findings from other studies and existing knowledge – are not explicitly validated with data in this study but are tested to better estimate their influences.

### 4.3.3 Sensitivity analysis and site-specific variations

The sensitivity analysis conducted with the snow module reveals significant site-specific variations in migration rates in response to ±10% changes in impact factors across the three study sites (CA, RU, AK). The higher sensitivity values observed at site CA likely reflect more extreme or variable environmental conditions, whereas the relatively lower sensitivity at site RU suggests a more stable environment where migration processes are less reactive to changes. Site AK exhibits high sensitivity



values, but the lack of statistically significant results in some scenarios suggests complex interactions and local factors not fully captured by the model.

The contrasting sensitivity observed in the scenario with a 10% increase in wind exposure highlights the complex role of wind in tree migration. At sites CA and AK, positive sensitivity values indicate that wind may enhance migration by aiding seed and pollen dispersal, which are crucial drivers of migration potential in these regions (Holtmeier and Broll, 2007; Kruse et al.,

2018). Wind-driven dispersal facilitates seed movement over long distances, likely explaining the positive association between wind exposure and migration at these sites (Holtmeier and Broll, 2007). However, the negative sensitivity observed at site RU suggests that wind exposure here may exert physiological stress that inhibits treeline migration. High wind velocities, for example, can damage trees through frost-drying and limit growth, a pattern documented in other wind-exposed treeline areas (Holtmeier, 1985). Moreover, seeds carried by strong winds tend to accumulate in sheltered areas, complicating migration in

exposed environments (Anschlag, 2006). Under such conditions, topographic factors such as wind exposure can outweigh potential dispersal benefits and prevent the advance of the treeline despite significant climate warming, as observed in other wind-exposed environments (Kullman, 2005; Beloiu et al., 2022). Additionally, wind erosion of exposed soils can reduce available seedbeds beyond the treeline, further inhibiting migration (Holtmeier et al., 2004; Anschlag, 2006). This duality underscores the complex role of wind in shaping treeline dynamics across varying environments.

Positive sensitivity to increased winter precipitation across all three study sites aligns with established ecological principles emphasising the role of moisture availability (Walker et al., 1999) and protective cover during critical early growth stages (Gross et al., 1991; Holtmeier, 2005a). Snow cover can provide crucial shelter for seedlings and saplings, protecting them from frost damage and herbivory during winter months, thereby supporting seedling survival (Juntunen and Neuvonen, 2006; Holtmeier and Broll, 2007). When snow melts, the resulting increase in soil moisture enhances seedling establishment during

the growing season, especially in regions where winter precipitation forms a significant portion of the annual water supply (Germino and Smith, 2000; Smith et al., 2003; Hagedorn et al., 2014). However, excessive snow accumulation can also have negative effects, such as snow creep or slides that may threaten seedling survival (Holtmeier, 2003; 2005a; 2005b; Kullman, 2005). This dual role of snow as both a protective factor and a potential hazard could explain why increased winter precipitation generally supports migration but may pose risks under extreme conditions (Holtmeier and Broll, 2007). Warmer temperatures,

combined with increased snowfall, as observed in some regions, can promote forest expansion but also lead to snow-related damage (Autio, 2006). These findings underscore how snow-rich environments can facilitate forest advancement by providing shelter and moisture during critical growth stages (Hagedorn et al., 2014; Kirdyanov et al., 2012; Devi et al., 2020; Grigoriev et al., 2022).

Enhanced facilitation by surrounding trees, as opposed to competition, has a strong positive effect on migration at sites CA

and RU. This likely arises from positive interactions within the growing tree population, where clusters of trees can moderate harsh conditions by creating warmer microclimates, dampening wind, and accumulating snowpack, thereby insulating young trees from extreme cold and herbivory (Germino and Smith, 2000; Germino et al., 2002; Holtmeier and Broll, 2007). Conversely, reduced wind exposure consistently shows a negative impact on migration at sites CA and RU. This could be due





to the important role wind plays in dispersing seeds across the landscape, as less wind exposure may reduce the ability of seeds
to travel efficiently, thereby hindering migration (Holtmeier and Broll, 2007). Although reduced wind exposure may alleviate
physiological stress on established trees, it can also delay or even prevent treeline advance by reducing seed dispersal in wind-
exposed environments where natural shelter is limited (Holtmeier, 2005a; Kullman, 2005).

The sensitivity analysis highlights the complex interactions between snow accumulation, wind exposure, and facilitation
processes in driving treeline migration. These processes are further shaped by site-specific factors such as topography and
climatic variability, which can either enhance or inhibit migration under changing environmental conditions. These findings
highlight the necessity of a comprehensive approach when assessing treeline migration in snow-dominated regions.

Our enhanced model represents a major advancement in simulating alpine treeline processes. By integrating snow effects on
seedling survival and soil conditions, alongside factors like wind exposure, facilitation by neighbouring trees, and
microtopography, the model provides a more comprehensive and accurate framework for understanding treeline migration in
high latitudes and elevations. This holistic approach is crucial for improving predictions and informing conservation strategies
in sensitive ecosystems. To refine our understanding, we challenged the model by incorporating various processes derived
from general models and previous studies to assess their effects. Although explicit validation was beyond the scope of this
study, this approach identifies key areas for future investigation to improve the accuracy of treeline dynamics modelling.

### 4.4 Constraints on treeline advance

The complexity of treeline migration is highlighted in our results, which show that site-specific climatic factors have the
strongest influence on migration rates, though the relative importance of these factors varies by location. This aligns with the
widely accepted view that the elevational position of treelines is climate-dependent in the absence of significant disturbance
regimes (Körner et al., 2020; Maher et al., 2021). However, our study underscores that treeline dynamics are not driven by
climate alone but also by additional environmental variables, such as local site conditions and the impacts of disturbances,
both natural and anthropogenic. These non-climatic influences complicate the relationship between climate warming and
treeline advance, as certain detrimental factors cannot be offset by even the positive effects of a warmer environment
(Holtmeier and Broll, 2005; 2007; 2019; Hansson et al., 2021).

A key finding of our research is the importance of snow-related processes, which have been incorporated into the LAVESI
snow module, in affecting tree migration rates within the treeline ecotone. Wiesner et al. (2019) suggest that, as mean surface
temperatures rise, the influence of extreme weather events (e.g., early or late frosts, or summer soil dryness) may become more
significant than gradual temperature increases. These extreme events, coupled with increasing snow cover due to climate
humidification, may act as a constraint on treeline advance in alpine regions.

Moreover, research on forest dynamics in high-elevation ecosystems highlights the need to consider both long-term warming
trends and climate variability, as these drive episodic dieback and tree invasions (Bugmann and Pfister, 2000; Milar et al.,
2004). While changes in climate averages are important, episodic events such as extreme cold episodes or changes in moisture





availability can profoundly impact treeline dynamics. These episodic effects have often been underrepresented in vegetation models.

This points to a potential explanation for the time lag between climate warming and observed treeline migration, as these short-term events and longer ecological processes complicate the response of tree populations to warming. The updated LAVESI version addresses this gap by explicitly accounting for the influence of varying snow depths on tree growth and treeline migration. By incorporating stochastically occurring extreme events, the model captures the complete spectrum of weather variability.

Another significant contributor to the time lag in treeline migration is the slow pace of population processes, such as seed dispersal, establishment, and growth. These slow processes can result in centuries-long delays between individual tree development and broader forest establishment (Lloyd, 2005; Holtmeier and Broll, 2007; 2019). Favourable conditions for seedling establishment may not align with good seed years, further delaying the upward shift of the treeline (Gruber et al., 2022). Despite these delays, over time, slow processes at the advancing forest edge will eventually bring tree populations into alignment with the climatic treeline and its fundamental niche (Körner, 2021). The persistence of relict populations from past warm periods, such as northernmost dwarf spruces in the Canadian tundra, further supports this idea of delayed treeline responses (Nichols, 1976; Holtmeier, 1985). Similarly, the legacy of past environmental conditions strongly shapes the distribution of plant communities in both the treeline ecotone and lower alpine regions (Hofgaard and Wilmann, 2002). The integration of snow processes into the LAVESI model enhances the accuracy of simulations related to seed dispersal, establishment, and growth, ultimately improving predictions of treeline advancement. However, the localised legacy effects of past populations vary significantly at small spatial scales. Addressing these variations requires further region-specific research to refine vegetation models and enhance their predictive capabilities.

Dial et al. (2022) propose that part of the delay in treeline advance may be due to conifers being on the brink of a stochastic, climate-driven invasion of tundra after centuries of stability. However, many treelines worldwide show slower upward shifts compared to the rate of densification, with tree populations rapidly increasing under favourable conditions (Kruse et al., 2016; Wang et al., 2016; Shi et al., 2022). The LAVESI model is capable of simulating not only treeline migration but also tree density within the specified simulation area. Additionally, shifts in reproductive methods – such as a move toward seedling-based regeneration in species like Engelmann spruce (*Picea engelmanii*) and subalpine fir (*Abies lasiocarpa*) – could facilitate greater dispersal distances and potentially accelerate treeline advancement in the future (Holtmeier and Broll, 2007). An upward shift of the alpine treeline in cold climate zones may also be limited by the fact that while short vegetation thrives in the relatively warmer microenvironment near the ground, trees can only reach full size if they can withstand the surrounding ambient temperatures (Wilson et al., 1987, Grace, 1989). Incorporating such characteristics that are partly species-specific and partly population-specific would further enhance the accuracy of vegetation models. Our findings underscore the complexity of treeline dynamics, suggesting that both climate factors, including snow-related processes, and slow ecological processes, such as seedling establishment, growth, survival, reproduction, and forest development, play crucial roles in shaping treeline



responses to climate change. The gaps highlight the need for further refinement of vegetation models to better explain the time
lag between climate warming and treeline advance.

## 5 Conclusions

In conclusion, this study highlights the distinct site-specific responses to factors influencing treeline shift and forest expansion, emphasising the critical role of localised conditions in shaping migration dynamics. While the findings reveal commonalities across sites, they also underscore unique interactions between environmental factors and local contexts. Notably, the responses
observed at the Canadian and the Russian sites provide clearer insights into the primary drivers of migration, whereas the high variability at the Alaskan site suggests more complex or less predictable local dynamics. This variability highlights the need for further investigation to reduce uncertainties in model predictions.

A key finding of this study is the significant role of snow in modulating migration potential, as snow accumulation creates favourable moisture conditions and protects seedlings from extreme cold, particularly in snow-dominated environments. These
results emphasise the importance of incorporating snow-related processes into vegetation models to improve predictions of boreal forest dynamics under changing environmental conditions.

Overall, this study enhances our understanding of tree migration processes and underscores the varied predictability of migration responses across different sites. These insights carry valuable implications for enhancing model accuracy and guiding conservation strategies aimed at sustaining alpine tundra resilience in the face of rapid environmental change.


*Code availability*. The source code of the host model is available from GitHub at https://github.com/StefanKruse/LAVESI/tree/circumboreal. The updated version presented here is named LAVESI-Mountain Treelines and the first version 1.0 is available for review on GitHub at https://github.com/StefanKruse/LAVESI/tree/circumboreal_mountain. Upon manuscript acceptance for publication, the code,
input data, and simulation output will be permanently archived on Zenodo.

*Author contributions*. Sarah Haupt: Conceptualisation; methodology; software; validation; formal analysis; investigation; resources; data curation; writing - original draft; writing - review & editing; visualisation. Josias Gloy: Software; data curation. Luca Farkas: Software; data curation; writing - review & editing. Katharina Schildt: Software; data curation; writing - review
& editing. Lisa Trimborn: Software; data curation; writing - review & editing. Stefan Kruse: Conceptualisation; methodology; software; validation; formal analysis; investigation; resources; data curation; writing - review & editing; supervision; project administration; funding acquisition.

*Competing interests*. The authors declare that they have no conflicts of interest.






*Acknowledgements*. We would like to thank Cathy Jenks for English language proofreading.

*Financial support*. The PhD position of Sarah Haupt was funded by the Russian–German DFG-project 'Climate-induced treeline dynamics in the Ural Mountains: drivers, constraints, and the role of genetic adaptation' (UralTreelines, https://gepris.dfg.de/gepris/projekt/448651799). We acknowledge support by the Open Access Publication Funds of Alfred-Wegener-Institut Helmholtz-Zentrum für Polar- und Meeresforschung.

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





# Appendix

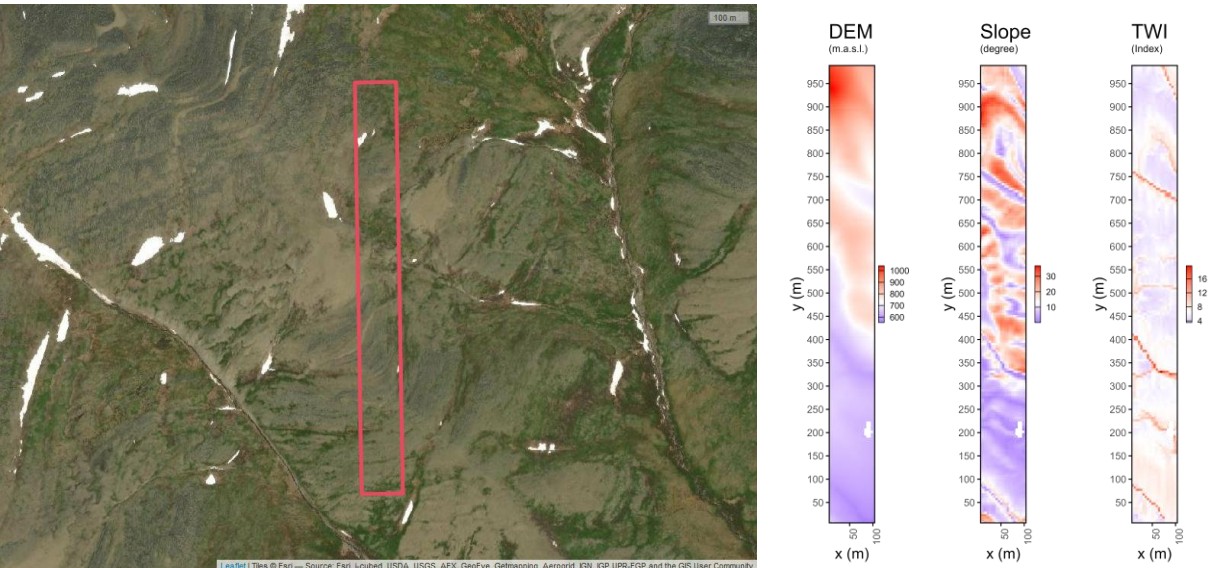

**Figure A1: Satellite image of the study site Fort McPherson, Canada (CA), including the simulation area (red box). Digital elevation model (DEM), slope, and topographic wetness index (TWI) for the simulation area.** Esri, i-cubed, USDA, USGS, AEX, Geoeye, Getmapping, Aerogrid, IGN, IGP, UPR-EGP, and the GIS User Community.

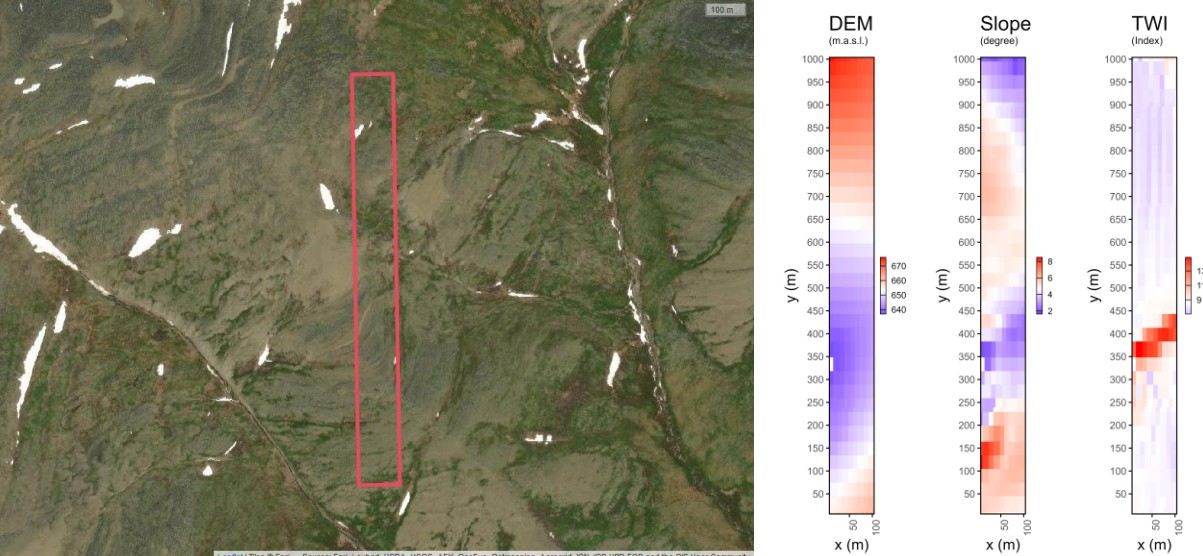

**Figure B1: Satellite image of the study site Lake Ilirney, Russia (RU), including the simulation area (red box). Digital elevation model (DEM), slope, and topographic wetness index (TWI) for the simulation area.** Esri, i-cubed, USDA, USGS, AEX, Geoeye, Getmapping, Aerogrid, IGN, IGP, UPR-EGP, and the GIS User Community.




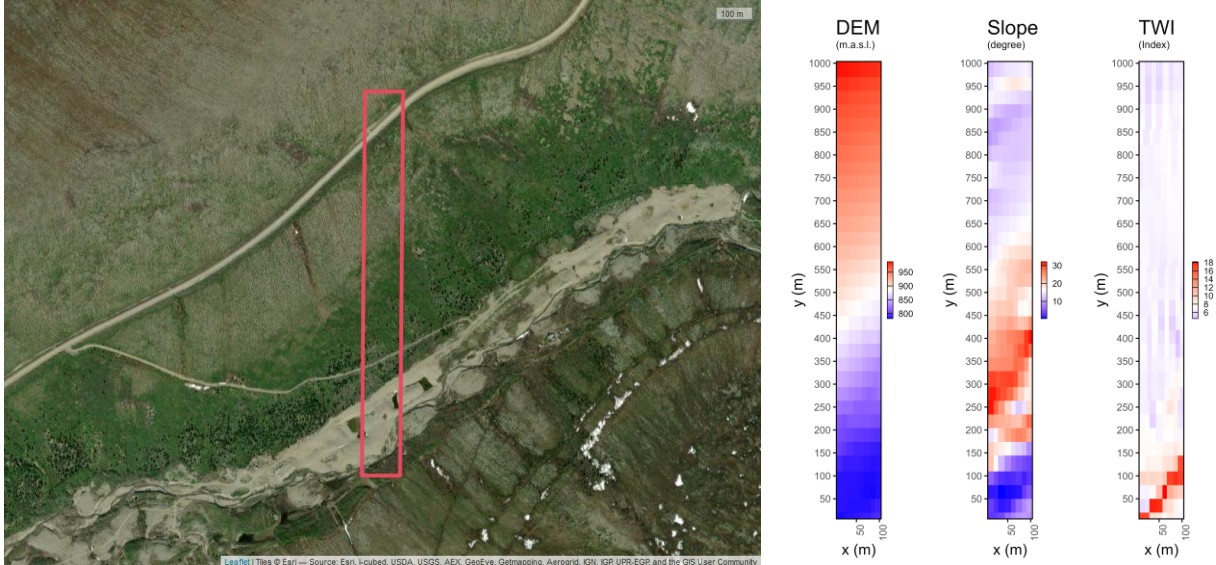

**Figure C1: Satellite image of the study site Road to Central, Alaska/ USA (AK), including the simulation area (red box). Digital elevation model (DEM), slope, and topographic wetness index (TWI) for the simulation area.** Esri, i-cubed, USDA, USGS, AEX, Geoeye, Getmapping, Aerogrid, IGN, IGP, UPR-EGP, and the GIS User Community.

**Table D1: Overview of simulation scenarios. Summary of the simulation scenarios and their respective multipliers for impact factors used in the sensitivity analysis.**

| Scenario ID | Scenario name | Impact factor | Multiplier |
|---|---|---|---|
| 0 | *reference* | *growingseasonlength* | 1.00 |
| | | *summertemperature* | 1.00 |
| | | *wintertemperature* | 1.00 |
| | | *summerprecipitation* | 1.00 |
| | | *windspeed* | 1.00 |
| | | *seedproduction* | 1.00 |
| | | *maturationage* | 1.00 |
| | | *seedintronumberpermanent* | 1.00 |
| | | *dispersaldistance* | 1.00 |
| | | *longdistancedispersal* | 1.00 |
| | | *seedlingestablishment* | 1.00 |
| | | *seedlingmortality* | 1.00 |
| | | *overageingmortality* | 1.00 |
| | | *slope* | 1.00 |
| | | *twi* | 1.00 |



| | | | |
|---|---|---|---|
| | | *seedbedavailability* | 1.00 |
| | | *activelayer* | 1.00 |
| | | *firemode* | 0.00 |
| | | *allow_pest_disturbances* | 0.00 |
| | | *pollination* | 0.00 |
| | | *snowcomputation* | 0.00 |
| 1 | *growingseasonlength_plus10%* | *growingseasonlength* | 1.10 |
| 2 | *growingseasonlength_minus10%* | *growingseasonlength* | 0.90 |
| 3 | *summertemperature_plus10%* | *summertemperature* | 1.10 |
| 4 | *summertemperature_minus10%* | *summertemperature* | 0.90 |
| 5 | *wintertemperature_plus10%* | *wintertemperature* | 1.10 |
| 6 | *wintertemperature_minus10%* | *wintertemperature* | 0.90 |
| 7 | *summerprecipitation_plus10%* | *summerprecipitation* | 1.10 |
| 8 | *summerprecipitation_minus10%* | *summerprecipitation* | 0.90 |
| 9 | *windspeed_plus10%* | *windspeed* | 1.10 |
| 10 | *windspeed_minus10%* | *windspeed* | 0.90 |
| 11 | *seedproduction_plus10%* | *seedproduction* | 1.10 |
| 12 | *seedproduction_minus10%* | *seedproduction* | 0.90 |
| 13 | *maturationage_plus10%* | *maturationage* | 1.10 |
| 14 | *maturationage_minus10%* | *maturationage* | 0.90 |
| 15 | *seedintronumberpermanent_plus10%* | *seedintronumberpermanent* | 1.10 |
| 16 | *seedintronumberpermanent_minus10%* | *seedintronumberpermanent* | 0.90 |
| 17 | *dispersaldistance_plus10%* | *dispersaldistance* | 1.10 |
| 18 | *dispersaldistance_minus10%* | *dispersaldistance* | 0.90 |
| 19 | *longdistancedispersal_plus10%* | *longdistancedispersal* | 1.10 |
| 20 | *longdistancedispersal_minus10%* | *longdistancedispersal* | 0.90 |
| 21 | *seedlingestablishment_plus10%* | *seedlingestablishment* | 1.10 |
| 22 | *seedlingestablishment_minus10%* | *seedlingestablishment* | 0.90 |
| 23 | *seedlingmortality_plus10%* | *seedlingmortality* | 1.10 |
| 24 | *seedlingmortality_minus10%* | *seedlingmortality* | 0.90 |
| 25 | *overageingmortality_plus10%* | *overageingmortality* | 1.10 |
| 26 | *overageingmortality_minus10%* | *overageingmortality* | 0.90 |
| 27 | *slope_plus10%* | *slope* | 1.10 |
| 28 | *slope_minus10%* | *slope* | 0.90 |





| 29 | *twi_plus10%* | *twi* | 1.10 |
|----|---------------|-------|------|
| 30 | *twi_minus10%* | *twi* | 0.90 |
| 31 | *seedbedavailability_plus10%* | *seedbedavailability* | 1.10 |
| 32 | *seedbedavailability_minus10%* | *seedbedavailability* | 0.90 |
| 33 | *activelayer_plus10%* | *activelayer* | 1.10 |
| 34 | *activelayer_minus10%* | *activelayer* | 0.90 |
| 35 | *firemode_ON* | *firemode* | 112.00 |
| 36 | *allow_pest_disturbances_ON* | *allow_pest_disturbances* | 1.00 |
| 37 | *pollination_ON* | *pollination* | 1.00 |
| 38 | *snowcomputation_ON_reference* | *snowcomputation* | 1.00 |
| | | *winterprecipitation* | 1.00 |
| | | *windexposure* | 1.00 |
| | | *facilitation* | 0.00 |
| 39 | *snowcomputation_ON_winterprecipitation_plus10%* | *snowcomputation* | 1.00 |
| | | *winterprecipitation* | 1.10 |
| 40 | *snowcomputation_ON_winterprecipitation_minus10%* | *snowcomputation* | 1.00 |
| | | *winterprecipitation* | 0.90 |
| 41 | *snowcomputation_ON_windexposure_plus10%* | *snowcomputation* | 1.00 |
| | | *windexposure* | 1.10 |
| 42 | *snowcomputation_ON_windexposure_minus10%* | *snowcomputation* | 1.00 |
| | | *windexposure* | 0.90 |
| 43 | *snowcomputation_ON_facilitation_plus10%* | *snowcomputation* | 1.00 |
| | | *facilitation* | 0.10 |
| 44 | *snowcomputation_ON_facilitation_minus10%* | *snowcomputation* | 1.00 |
| | | *facilitation* | -0.10 |