# Peer review of "The significant role of snow in shaping alpine treeline responses in modelled boreal forests"

_EGUsphere, 2024_

## Author Comment (AC2)

**Citation**: https://doi.org/10.5194/egusphere-2024-4036-RC2

RC2: The manuscript entitled "The significant role of snow in shaping alpine treeline responses in modelled boreal forests" aims to investigate the effect of snow on the treeline shift. However, in my opinion, reading the article looks like this is not the main focus of the manuscript. The structure of the article is center on the sensitivity analysis of the forest dynamic model LAVESI. In the methodology section is reported the description of a new way to incorporate the effect of the snow cover in the LAVESI model, but the results section is just partially focused on the outcomes of this process. Some issues in the incorporation of this process are detected and reported in the specific comments below. Moreover, the study areas are not adequately described and neither pictures or orthophoto are reported. For me it is complicate to understand the real characteristics of the three investigated sites.

*AC: Thank you for your insightful feedback and constructive critique during the first review of our preprint. Your comments provide important guidance for improving our manuscript. We will carefully consider all referees' suggestions to enhance the clarity and quality of our study, ensuring it aligns with the high standards of Biogeosciences.*

RC2: The approach adopted in this manuscript is the progressive variation of the model inputs and so it is basically a sensitivity analysis of the LAVESI model.

*AC: We did not conduct a progressive variation of the model inputs but, as correctly noted, performed a sensitivity analysis. In the context of pattern-oriented modeling, this is a standard approach in ecological individual-based modeling (see Grimm & Railsback, 2005, Individual-based Modeling and Ecology, Princeton University Press; Grimm et al., 2005, Science, 310: 987-991).*

RC2: Furthermore, there is no comparison between the model outputs and the actual forest characteristics making impossible to evaluate the reliability of the model outcomes and consequently the future scenarios.

*AC: In pattern-oriented modeling, a simple model is iteratively refined to more accurately reproduce observed patterns. As part of the validation process, model outputs are compared with actual forest characteristics. While we have fitted the treelines, this was not done in a spatially explicit manner. The original LAVESI model (Kruse et al., 2016) has been systematically extended to incorporate dispersal processes influenced by wind speed and direction (LAVESI-WIND 1.0; Kruse et al., 2018), landscape topography and additional boreal forest species (LAVESI-CryoGrid v1.0; Kruse et al., 2022), and forest fire dynamics (LAVESI-FIRE; Glückler et al., 2024). Both the pattern-oriented modeling approach and the sensitivity analysis have been consistently applied across these studies.*

RC2: The manuscript need to be deeply restructured and also the methodological approach deeply revised. In my opinion the final suggestion for the manuscript is a rejection encouraging the authors to work on it for a resubmission.

*AC: As also pointed out in Referee Comment 1, we will revise and clarify the description while maintaining the methodology, as it aligns with established ecological modeling standards.*

RC2: Below I report the specific comments.

Introduction: the section is too long and I suggest the authors to reduce it of about 50 lines. Moreover a general comment to this section is to restructure the description of the drivers and factors influencing the treeline shift. As it is, in my opinion it results a little bit complete to follow.

*AC: We will shorten the introduction while preserving all essential content. Currently, the influencing factors are categorized into abiotic climatic factors, species-dependent factors, abiotic non-climatic factors, and other biotic factors. We will further emphasize this classification and illustrate it with a graphical overview.*

RC2: Line 187: for run, the authors intended years? So, the model time resolution is equal to 1 year if I understood correctly. Please specify it.

*AC: The model description outlines that within one simulation step, which is one year, the relevant processes are incorporated consecutively as submodules (see lines 175-176). We will reiterate this in line 187 for clarity.*

RC2: Section 2.1: how do the species-specific parameters included in the model were derived? Please refer also to the species you are going to simulate reported in table 1.

*AC: We will compile a complete table of species traits and model variable values in LAVESI and include it in the manuscript appendix. We will provide a clearer explanation of the site-specific results by directly linking environmental factors and species traits to site-specific variability, thereby enhancing the understanding of the observed differences across sites.*

RC2: Line 194: How does the tree mortality rate is included in the model? Please specify what tree and environmental related parameters where used to model this process.

*AC: The fundamental methodology for determining seed and tree mortality was originally developed by Kruse et al. (2016) and has been updated since. We will provide a clear explanation of how the tree mortality rate was calculated, detailing the environmental parameters that influence this process and the mechanisms behind their effects.*

RC2: Line 221: the threshold of 5°C for distinguish between rainfall and snowfall needs to be accurately motivate. The value of 5 sounds quite high in my opinion. How do you motivate it? Temperature daily fluctuation in case of precipitation is really low so I would expect that a mean daily temperature slightly less than 5°C will lead to rainfall precipitation. This is a crucial point, and it needs to be clarify and accurately motivated since it can majorly affect the simulation results.

*AC: Our approach is based on the empirical function from Sato et al. (2007), which in turn refers to the precipitation partitioning approach from Ichii et al. (2003). Ichii et al. (2003) used a gradual transition, where snowfall accounts for 50% of total precipitation at 2°C. In our model development, we initially incorporated this approach but subsequently adjusted the threshold as part of a model tuning process. Specifically, we compared simulated and observed snow accumulation patterns and iteratively refined the temperature threshold to improve model accuracy. However, we acknowledge that this value may vary depending on regional climate conditions, and we will clarify this tuning process more explicitly in the revised manuscript.*

RC2: Line 245-250: about avalanche induced mortality, the authors never mentioned the role of the slope as predisposing factor for snow avalanche release. It is generally defined that a slope between 27° and 55° is the range where avalanches can occur.

*AC: We implemented snow movements downslope in extreme years which happen with a chance increasingly when exceeding the 99.9th percentile of snow height in the full climate forcing series. This movement is simply implemented as downslope independent from the slope angle. Further, in this simplification we named this process avalanche. In the snow module description, we then described the process and stated that mortality is added where a modelled avalanche impacts the leading edge of tree stands, with its destructive force diminishing downslope due to protecting tree growth (see lines 250-251). We will provide a more detailed explanation that snow accumulation and snow movement were modeled for each grid cell, considering the slope direction.*

RC2: Section 2.3: For me this section results slightly complicate to follow because it is a pure description of the model and the relative processes. I would encourage the authors to use formulas and/or diagrams or flowcharts in order to have also a clear view of the developed model.

*AC: We will include additional formulas where needed and in-text formulas are not sufficient. We will create a flowchart to clarify the interrelated processes. Additionally, a graphical overview of all influencing factors will be provided to facilitate a better understanding of the overall framework.*

RC2: Table 1: are the locations reported in a local or a geographic coordinate system? It looks to me east north but the xmin etc. is confusing me.

*AC: We will update the names to 'Longitude' and 'Latitude' for clarity.*

RC2: I would suggest to add also the total area modelled for each study site.

*AC: We will also specify the total simulation area for each study site at this point.*

RC2: A small map would be beneficial to clearly identify the location of the investigated sites.

*AC: As requested, we will include a map to visually illustrate the locations of the study sites.*

RC2: Lines 304-308: Why do the authors compute the sensitivity analysis of the parameters of the model? For me it is unclear the motivation of this operation.

*AC: As previously mentioned, this approach is a standard method in ecological modeling (Grimm & Railsback, 2005; Grimm et al., 2005). The methods section further explains that the sensitivity analysis is conducted to evaluate and compare the influence of various factors on the migration rate of the alpine treeline and to identify key drivers and assess how variations in these factors influence treeline dynamics (see lines 299-300).*

RC2: Section 5: for really understand the morphology of the sites and the condition it would be beneficial to add an image with the orthophoto of the investigated areas with contour lines or something similar.

*AC: Satellite images of the study sites are already included in the preprint. However, we have noticed that Figures A1 and B1 mistakenly display the same image, which we will correct. We initially decided against including these images in the main section to avoid excessive length but are happy to move them if desired. Additionally, we can provide photos and full panorama images of the field plots online.*

RC2: Section 5: It would be of great value if the authors can compare the actual condition of the study site with the results of the spin up simulation. In this way it is possible to assess the reliability of the model and therefore the validity of the future scenarios.

AC: *The treeline was adjusted to match the current observed position through a fitting process, which, although conducted, has not yet been explained in detail within the manuscript. We will address this more thoroughly in the revised version. However, it is important to note that achieving an exact match in a stochastic model is inherently challenging. The model generates mean positions based on multiple runs, with some simulations advancing further while others lag behind. This variability is a result of the stochastic nature of the model, and it allows us to explore the associated uncertainties. For further clarification, please refer to the section on stochasticity in Grimm & Railsback (2005).*

RC2: Line 330: What is the definition of treeline? It is necessary to define how the authors have identified it. Is it based on a certain density of trees or something similar?

*AC: The definition of the treeline in this study is described in lines 328-329: Treelines were determined using threshold-based criteria of one hundred trees per hectare.*

RC2: General comment for the methodology section: Authors investigated which factors are the most important/influent for the migration rate. I would say that the common approach for simulating future scenarios is to have a solid and reliable model either tested with field data or from literature in similar conditions. Consequently, models are used to predict future change by just modifying the environmental variables or the initial conditions, obtaining in this case the probable shift of the treeline. In this manuscript the procedure is not classical, and the authors look like they want to investigate which variable of the model can affect the treeline shift and how. To me this approach is more similar to a simple exercise respect to really understand the possible dynamic of the treeline shift for the three study sites. This feeling is corroborated by the absence of comparison between the current condition of the forests and simulation spin up for the three study areas. In other word the factors influencing the tree line shift are the environmental one and not the parameters of the model.

*AC: The aim of our study and sensitivity analysis was to evaluate and compare the influence of various factors on the migration rate of the alpine treeline, identify key drivers, and assess how variations in these factors influence treeline dynamics. As outlined by Grimm & Railsback (2005) and Grimm et al. (2005), this approach is a standard method for determining the influence of input factors on model output. This methodology has also been applied in several other LAVESI studies, such as Kruse et al. (2016, 2018, 2022) and Glückler et al. (2024). Similar approaches have been used in other studies, including Zurell et al. (2011), who linked species distribution models with an individual-based model to assess climate-induced*

***range shifts in black grouse, and [Malchow et al. (2024)](), who used a spatially
explicit individual-based model to analyze red kite population dynamics.
Additionally, [Zurell et al. (2018)]() applied species distribution models to evaluate the
risks faced by migratory birds due to climate and land cover changes. These
studies demonstrate that integrating process-based models with sensitivity
analyses is a widely used method for understanding ecological dynamics under
environmental change.***

RC2: Line 359: please add a couple of sentences to introduce the results instead of
directly showing Table 3.

***AC: In accordance with the suggestion, we will add an introductory sentence at
this point.***

RC2: Table 5: The table should be moved to the results section.

***AC: As suggested, we will move Table 5 to the results section.***

RC2: Lines 458-462: I would say this is a quite obvious result. The study areas are
different, and I would expect exactly this result.

***AC: We will provide a more detailed explanation of the site-specific differences in
the sensitivity analysis results, thereby clarifying or reassessing the significance
of this section.***

RC2: General comment to the results and discussion section: The authors report the
outcomes of the modelled scenarios discussing and comparing the relative results, but
the real problem is that a reliable scenario has not been identified. In the manuscript
there is no identification of a modelled scenario that can be representative of
current/actual forest conditions and of the future one. Without this definition is not
possible to quantify the alpine treeline shift.

***AC: We appreciate the reviewer's concern regarding the identification of a
representative scenario for current and future forest conditions. However, our
study is not designed to provide deterministic predictions of future treeline
positions. Instead, it is a sensitivity study aimed at understanding the relative
influence of different factors on treeline migration dynamics. Our model was tuned
to represent observed patterns, but the primary objective is to assess the key
drivers and interactions shaping treeline shifts rather than to predict an exact
future distribution. By systematically varying input factors, we can identify the
processes most relevant for treeline migration, which can help refine future
projections and improve ecological understanding rather than provide a single
"correct" scenario. We will clarify this point more explicitly in the manuscript.***

---

## Author Response (AR1)

Citation: https://doi.org/10.5194/egusphere-2024-4036-RC1

RC1: This article is investigating the role of snow on treeline response to climatic warming by including snow (in addition to other abiotic and biotic variables) module in the vegetation model LAVESI. I like the idea with the model LAVESI. Results showed site specific response of treeline shifts and highlighted the role of snow in treeline dynamics. The approach including additional variable in vegetation model is interesting. The data are enough and relevant to test the authors' hypothesis. I recommend a minor revision.

AC: We sincerely appreciate your valuable feedback and constructive comments during the first review of our preprint. Your comments significantly enhanced our manuscript. We carefully addressed all referees' suggestions to further improve the quality of our study and ensure it meets the high standards of Biogeosciences.

RC1: I like your introduction, but it is a bit too long, it would very nice if you are able to make it more concise by reducing it by 10-20 % pages.

AC: As recommended, we shortened the introduction while preserving all essential content.

RC1: Recently several studies reported role of biotic interactions on treeline dynamics. Please see Bader et al. 2021, Ecography, Liang et al. 2016, PNAS, Sigdel et al. 2024, Nature Plants.

AC: Thank you for the additional literature suggestions. We included them where fitting in the text (Introduction, Discussion).

RC1: Overall, the discussion is general and comprehensive for me and mostly authors focused on those influences and processes that have been investigated. It has been neglected role of snow duration, i.e., the period when snow covers the treeline, determines processes of tree regeneration and establishment.

AC: The snow module accounts for snow duration, i.e., the period during which snow covers the treeline area:

Building on the approach of Sato et al. (2007), the processes of snow accumulation and snowmelt were incorporated into the LAVESI model (see lines 231-232). As a result, the duration of snow cover is influenced by snow depth, which is affected by both snow accumulation and snowmelt. All subsequent influences on tree growth and treeline migration, which are dependent on snow depth, are consequently impacted throughout the duration of the snow cover. The effects of varying snow depths were implemented in the model in the following abstracted manner:

If the height of the tree is higher than the current snow depth, the mortality rate of the tree increases by 50%. This penalty decreases linearly with increasing tree height and is eliminated once the tree reaches a height of 5 m.

When seeds are covered by snow, their germination is enhanced, exhibiting a linear increase in germination rates with greater snow depth. Specifically, this relationship begins at zero and reaches a 30% increase in germination at a snow depth of 1 m.

If seedlings and saplings are covered by snow, their growth is enhanced by 30%. The germination process is influenced linearly, either positively or negatively, based on whether the snow melts before or after the 110th day of the year.

The actual day of snowmelt is divided by 110, and the deviation from 1.0 is added to the germination probability (see lines 237-243).

In general, the growing season length and its influence on seedling germination and tree growth are already incorporated into the vegetation model (see lines 187-189).

In the revised version of the manuscript, we have thoroughly revised the description of the newly implemented snow module and included a flow chart to illustrate its structure and processes more clearly.

RC1: Related to this, snow fungi might also explain part of the spatio-temporal pattern.

AC: The influence of snow fungi was indirectly incorporated into the snow module, as prolonged high snow load or a shortened growing season negatively affects tree growth. In the revised version of the manuscript, we have expanded the discussion to acknowledge that snow fungi may also contribute to the observed spatio-temporal pattern.

RC1: Also snow movements and avalanches may affect the observed species differently.

AC: The snow module includes snow load and avalanche occurrence. However, we did not vary these effects by tree species, as the simulation areas in our study – as well as the North American and Siberian treeline ecotones in general – are predominantly dominated by spruces in Northwest Canada and Alaska, and by larches in Siberia.

In the revised version of the manuscript, we included species-dependent differences in treeline migration in the discussion to provide further insights for studies focusing on species-specific responses. We focus in particular on the differences in species composition at the three study sites in the reference run, without delving into further detail on the other scenarios.

RC1: Additionally, even though, different plant traits and interactions are included in the model, their influence on treeline sensitivity and migration is missing.

AC: Species-dependent plant traits are incorporated into the vegetation model. We acknowledge that these species-specific traits can contribute to differences in treeline responses; however, as our results demonstrate, these effects are outweighed by the climatic differences among locations. This is further addressed in the revised version of the manuscript.

RC1: It is not clear why site-specific specifics sensitivities are observed? What are the mechanisms behind such discrepancies? It needs to be justified based on site specific evidences. Authors discussed different factors (environmental and traits) but fail to link site specific variability. Thus, it needs to synthesize the results rather than presenting results directly.

AC: We acknowledge the need for a more integrated discussion of site-specific sensitivities. Rather than treating the study sites separately, in the revised version of the manuscript, we synthesized the results to highlight overarching patterns and mechanisms driving the observed differences. By linking environmental factors and species traits more explicitly, we aim to provide a clearer, more comprehensive understanding of the site-specific variability in treeline dynamics.

RC1: Generally, authors just compare their results with other similar studies. To make it convincing, it needs deeper discussion with key scientific evidence and how it helps to advance our understanding on treeline ecology under changing climate.

AC: In the discussion of the revised manuscript, we highlighted the implications of our results for the future of North America's and Siberia's treelines in the context of global warming.

RC1: Also, ecological implications of the modified model should be highlighted.

AC: In the discussion of the revised manuscript, we also further elaborated on the ecological implications for North American and Siberian forests, including biodiversity, the albedo effect, and their role as a carbon sink.

RC1: L13-14: As site wise results are presented, it is better to mention the areas/sites considered in analysis.

AC: In the revised version of the manuscript, we mentioned the study sites at this point in the abstract.

RC1: L64-66: Please see recent treeline studies used individual-based model to simulate plot-based data including both biotic and abiotic variables (Zheng et al. 2024, Eco Lett).

AC: Thank you for the literature suggestion. We carefully reviewed the methods and findings of this study.

RC1: L184: Please mention the sites considered in this study before this sentence.

AC: In the revised version of the manuscript, we introduced the study sites earlier in the manuscript.

RC1: L307-308: Even though the trees traits data were adopted from previous publications, better to elaborate how these data were retrieved and sources.

AC: For the revised version of the manuscript, we compiled a complete table of species traits and model variable values in LAVESI and included it in the manuscript supplement.

RC1: Table 3 caption: impact factors could be replaced by predicting variables.

AC: We changed this term according to your suggestion.

RC1: No any parameter is significance from Road to central Alaska. What is mean? Is this model not well able to predict the sensitivity of Alaskan treelines?

AC: At the Road to Central, Alaska site, no single factor was found to be statistically significant. This may be due to stronger interactions between multiple factors, which could obscure the effects of individual parameters. Another possible explanation is that this site is more influenced by stochastic processes. This aspect is now addressed in greater detail in the revised discussion.

RC1: L392-395: It seems to methodology rather than results.

AC: In the revised version of the manuscript, this section has been adapted to function as an introduction to the results rather than a reiteration of the methods.

RC1: L464-469: Just speculations, no evidence supports the findings.

AC: As mentioned above, we provided a more detailed explanation of the sitespecific differences in the sensitivity analysis results, thereby clarifying or reassessing the significance of this section.

RC1: L532: either data or reference should be presented.

AC: In the revised version of the manuscript, we inserted a reference to 'Table 1: Characteristics of the study sites.' at this point.

RC1: L550: How did species-specific stand structures varies across the sites. It should be site and species specific. How population structure varied across the study sites? and so on.

AC: Since spruce or larch clearly dominates at all three study sites, we have not yet conducted a detailed comparison of species composition. However, we included a general comparison of the stand structure of the study sites in the revised version of the manuscript.

Citation: https://doi.org/10.5194/egusphere-2024-4036-RC2

RC2: The manuscript entitled "The significant role of snow in shaping alpine treeline responses in modelled boreal forests" aims to investigate the effect of snow on the treeline shift. However, in my opinion, reading the article looks like this is not the main focus of the manuscript. The structure of the article is center on the sensitivity analysis of the forest dynamic model LAVESI. In the methodology section is reported the description of a new way to incorporate the effect of the snow cover in the LAVESI model, but the results section is just partially focused on the outcomes of this process. Some issues in the incorporation of this process are detected and reported in the specific comments below. Moreover, the study areas are not adequately described and neither pictures or orthophoto are reported. For me it is complicate to understand the real characteristics of the three investigated sites.

AC: Thank you for your insightful feedback and constructive critique during the first review of our mansucript. Your comments provide important guidance for improving our manuscript. We carefully considered all referees' suggestions to enhance the clarity and quality of our study, ensuring it aligns with the high standards of Biogeosciences.

We had indeed intended this study as a sensitivity analysis to explore the effect sizes and relative significance of various processes, specifically those arising from the inclusion of snow in a spatially explicit model. In response to the critiques raised, we have carefully addressed each point in the specific comments section below.

Satellite images of the study sites are already included in the preprint.

Furthermore, we believe that the original title accurately captures the essence of the study's objective, and as such, we propose to retain it. The revised manuscript, which now incorporates all recommended edits and suggestions from both reviewers, should make this intention clearer.

RC2: The approach adopted in this manuscript is the progressive variation of the model inputs and so it is basically a sensitivity analysis of the LAVESI model.

AC: We did not conduct a progressive variation of the model inputs but, as correctly noted, performed a sensitivity analysis. We conducted reference simulations and subsequently varied individual predictive variables. In the context of pattern-oriented modeling, this is a standard approach in ecological individual-based modeling (see Grimm & Railsback, 2005, Individual-based Modeling and Ecology, Princeton University Press; Grimm et al., 2005, Science, 310: 987-991).

RC2: Furthermore, there is no comparison between the model outputs and the actual forest characteristics making impossible to evaluate the reliability of the model outcomes and consequently the future scenarios.

AC: In pattern-oriented modeling, a simple model is iteratively refined to more accurately reproduce observed patterns. As part of the validation process, model outputs are compared with actual forest characteristics. While we have fitted the treeline position on the mountain slopes in general terms, this was not done in a spatially explicit manner. Prior to this study, the original LAVESI model (Kruse et al., 2016) has been systematically extended to incorporate dispersal processes

influenced by wind speed and direction (LAVESI-WIND 1.0; Kruse et al., 2018), landscape topography and additional boreal forest species (LAVESI-CryoGrid v1.0; Kruse et al., 2022), and forest fire dynamics (LAVESI-FIRE; Glückler et al., 2024). Both the pattern-oriented modeling approach and the sensitivity analysis have been consistently applied across these studies. In the revised manuscript, we have cited the relevant literature on which this approach is based.

RC2: The manuscript need to be deeply restructured and also the methodological approach deeply revised. In my opinion the final suggestion for the manuscript is a rejection encouraging the authors to work on it for a resubmission.

AC: As also pointed out in Referee Comment 1, we revised and clarified the description while maintaining the methodology, as it aligns with established ecological modeling standards.

RC2: Below I report the specific comments.

Introduction: the section is too long and I suggest the authors to reduce it of about 50 lines. Moreover a general comment to this section is to restructure the description of the drivers and factors influencing the treeline shift. As it is, in my opinion it results a little bit complete to follow.

AC: In the revised version of the manuscript, we shortened the introduction while preserving all essential content. Now, the influencing factors are categorized into abiotic climatic factors, abiotic non-climatic factors, species-dependent factors, and other biotic factors. We further emphasized this classification and illustrate it with a graphical overview.

RC2: Line 187: for run, the authors intended years? So, the model time resolution is equal to 1 year if I understood correctly. Please specify it.

AC: The model description outlines that within one simulation step, which is one year, the relevant processes are incorporated consecutively as submodules (see lines 175-176). In the revised version of the manuscript, we reiterated this in line 187 for clarity.

RC2: Section 2.1: how do the species-specific parameters included in the model were derived? Please refer also to the species you are going to simulate reported in table 1.

AC: For the revised version of the manuscript, we compiled a complete table of species traits and model variable values in LAVESI and include it in the manuscript supplement. We provided a clearer explanation of the site-specific results by directly linking environmental factors and species traits to site-specific variability, thereby enhancing the understanding of the observed differences across sites.

RC2: Line 194: How does the tree mortality rate is included in the model? Please specify what tree and environmental related parameters where used to model this process.

AC: The fundamental methodology for determining seed and tree mortality was originally developed by Kruse et al. (2016) and has been updated since. In the revised version of the manuscript, we provided a clear explanation of how the tree

mortality rate was calculated, detailing the environmental parameters that influence this process and the mechanisms behind their effects.

RC2: Line 221: the threshold of 5°C for distinguish between rainfall and snowfall needs to be accurately motivate. The value of 5 sounds quite high in my opinion. How do you motivate it? Temperature daily fluctuation in case of precipitation is really low so I would expect that a mean daily temperature slightly less than 5°C will lead to rainfall precipitation. This is a crucial point, and it needs to be clarify and accurately motivated since it can majorly affect the simulation results.

AC: While our study did not employ a 5°C temperature threshold, this value was used by Forster et al. (2017) in their approach.

Our approach is based on the empirical function from Sato et al. (2007), which in turn refers to the precipitation partitioning approach from Ito and Oikawa (2002). Ito and Oikawa (2002) used a gradual transition, where snowfall accounts for 50% of total precipitation at 2°C. In our model development, we initially incorporated this approach but subsequently adjusted the threshold as part of a model tuning process. Specifically, we compared simulated and observed snow accumulation patterns and iteratively refined the temperature threshold to improve the simulated snow depth accuracy. However, we acknowledge that this value may vary depending on regional climate conditions, and we clarified this tuning process more explicitly in the revised manuscript.

RC2: Line 245-250: about avalanche induced mortality, the authors never mentioned the role of the slope as predisposing factor for snow avalanche release. It is generally defined that a slope between 27° and 55° is the range where avalanches can occur.

AC: We implemented snow movements downslope in extreme years which happen with a chance increasingly when exceeding the 99.9th percentile of snow height in the full climate forcing series. This movement is simply implemented as downslope independent from the slope angle. Further, in this simplification we named this process avalanche. In the snow module description, we then described the process and stated that mortality is added where a modelled avalanche impacts the leading edge of tree stands, with its destructive force diminishing downslope due to protecting tree growth (see lines 250-251). We provided a more detailed explanation that snow accumulation and snow movement were modeled for each grid cell, considering the slope direction. Indeed, with our approach, we appear to overestimate snow movement, especially when it is restricted to steeper slope angles. However, the impact of this overestimation still seems to be relatively minor. Future versions of this model could improve upon this by testing it further on slopes more prone to avalanches.

RC2: Section 2.3: For me this section results slightly complicate to follow because it is a pure description of the model and the relative processes. I would encourage the authors to use formulas and/or diagrams or flowcharts in order to have also a clear view of the developed model.

AC: In the revised version of the manuscript, we included formulas where needed and in-text formulas are not sufficient. We created a flowchart to clarify the interrelated processes. Additionally, we provide a graphical overview of all influencing factors to facilitate a better understanding of the overall framework.

RC2: Table 1: are the locations reported in a local or a geographic coordinate system? It looks to me east north but the xmin etc. is confusing me.

AC: We updated the names to 'Longitude' and 'Latitude' for clarity.

RC2: I would suggest to add also the total area modelled for each study site.

AC: In the revised version of the manuscript, we also specified the total simulation area for each study site at this point.

RC2: A small map would be beneficial to clearly identify the location of the investigated sites.

AC: As requested, we included a map to visually illustrate the locations of the study sites.

RC2: Lines 304-308: Why do the authors compute the sensitivity analysis of the parameters of the model? For me it is unclear the motivation of this operation.

AC: As previously mentioned, this approach is a standard method in ecological modeling (Grimm & Railsback, 2005; Grimm et al., 2005). The methods section further explains that the sensitivity analysis is conducted to evaluate and compare the influence of various factors on the migration rate of the alpine treeline and to identify key drivers and assess how variations in these factors influence treeline dynamics (see lines 299-300 in the preprint).

RC2: Section 5: for really understand the morphology of the sites and the condition it would be beneficial to add an image with the orthophoto of the investigated areas with contour lines or something similar.

AC: Satellite images of the study sites are already included in the preprint. However, we have noticed that Figures A1 and B1 mistakenly display the same image, which we corrected. We initially decided against including these images in the main section to avoid excessive length but are happy to move them. Additionally, we provided photos of the study sites.

RC2: Section 5: It would be of great value if the authors can compare the actual condition of the study site with the results of the spin up simulation. In this way it is possible to assess the reliability of the model and therefore the validity of the future scenarios

AC: The treeline was adjusted to match the current observed position through a fitting process, which, although conducted, has not yet been explained in detail within the manuscript. We addressed this more thoroughly in the revised version. However, it is important to note that achieving an exact match in a stochastic model is inherently challenging. The model generates mean positions based on multiple runs, with some simulations advancing further while others lag behind. This variability is a result of the stochastic nature of the model, and it allows us to explore the associated uncertainties. For further clarification, please refer to the section on stochasticity in Grimm & Railsback (2005).

RC2: Line 330: What is the definition of treeline? It is necessary to define how the authors have identified it. Is it based on a certain density of trees or something similar?

AC: The definition of the treeline in this study is described in lines 328-329: Treelines were determined using threshold-based criteria of one hundred trees per hectare.

RC2: General comment for the methodology section: Authors investigated which factors are the most important/influent for the migration rate. I would say that the common approach for simulating future scenarios is to have a solid and reliable model either tested with field data or from literature in similar conditions. Consequently, models are used to predict future change by just modifying the environmental variables or the initial conditions, obtaining in this case the probable shift of the treeline. In this manuscript the procedure is not classical, and the authors look like they want to investigate which variable of the model can affect the treeline shift and how. To me this approach is more similar to a simple exercise respect to really understand the possible dynamic of the treeline shift for the three study sites. This feeling is corroborated by the absence of comparison between the current condition of the forests and simulation spin up for the three study areas. In other word the factors influencing the tree line shift are the environmental one and not the parameters of the model.

AC: The aim of our study and sensitivity analysis was to evaluate and compare the influence of various factors on the migration rate of the alpine treeline, identify key drivers, and assess how variations in these factors influence treeline dynamics. As outlined by Grimm & Railsback (2005) and Grimm et al. (2005), this approach is a standard method for determining the influence of input factors on model output. This methodology has also been applied in several other LAVESI studies, such as Kruse et al. (2016, 2018, 2022) and Glückler et al. (2024). Similar approaches have been used in other studies, including Zurell et al. (2011), who linked species distribution models with an individual-based model to assess climate-induced range shifts in black grouse, and Malchow et al. (2024), who used a spatially explicit individual-based model to analyze red kite population dynamics. Additionally, Zurell et al. (2018) applied species distribution models to evaluate the risks faced by migratory birds due to climate and land cover changes. These studies demonstrate that integrating process-based models with sensitivity analyses is a widely used method for understanding ecological dynamics under environmental change.

RC2: Line 359: please add a couple of sentences to introduce the results instead of directly showing Table 3.

AC: In accordance with the suggestion, we added an introductory sentence at this point.

RC2: Table 5: The table should be moved to the results section.

AC: As this table comparing expected and observed results relates directly to the discussion, we prefer to retain it in its current position.

RC2: Lines 458-462: I would say this is a quite obvious result. The study areas are different, and I would expect exactly this result.

AC: In the revised version of the manuscript, we provided a more detailed explanation of the site-specific differences in the sensitivity analysis results, thereby clarifying or reassessing the significance of this section.

RC2: General comment to the results and discussion section: The authors report the outcomes of the modelled scenarios discussing and comparing the relative results, but the real problem is that a reliable scenario has not been identified. In the manuscript there is no identification of a modelled scenario that can be representative of current/actual forest conditions and of the future one. Without this definition is not possible to quantify the alpine treeline shift.

AC: We appreciate the reviewer's concern regarding the identification of a representative scenario for current and future forest conditions. However, our study is not designed to provide deterministic predictions of future treeline positions. Instead, it is a sensitivity study aimed at understanding the relative influence of different factors on treeline migration dynamics. Our model was tuned to represent observed patterns, but the primary objective is to assess the key drivers and interactions shaping treeline shifts rather than to predict an exact future distribution. By systematically varying input factors, we can identify the processes most relevant for treeline migration, which can help refine future projections and other modelling approaches and improve ecological understanding rather than provide a single "correct" scenario. We clarified this point more explicitly in the revised version of the manuscript.

---

## Author Response (AR2)

**Subject: Cover Letter for 2nd Resubmission of Manuscript**

Dear Prof. Dr. Garbarino, Dr. Hagedorn, Dr. Migliavacca and reviewers,

I am pleased to submit the revised manuscript, entitled 'The significant role of snow in shaping alpine treeline responses in modelled boreal forests' for your careful review and consideration for publication in your special issue on the interactions between abiotic and biotic factors shaping vegetation patterns in treeline ecotones.

I wish to express my sincere gratitude for your continued feedback and constructive comments during the review of the revised version of our manuscript. We have carefully addressed each of the reviewers' comments and recommendations, which has further improved the quality of our study. I am confident that the revised manuscript is now even more in line with the high standards set by *Biogeoscience*.

We are looking forward to your evaluation and the possibility of having the revised manuscript accepted for publication in *Biogeoscience*. Please feel free to contact me should you require any further clarification or information.

Thank you for your time and consideration.

Sincerely, on behalf of all authors, Sarah Haupt

Polar Terrestrial Environmental Systems
Alfred-Wegener-Institute, Helmholtz Centre for Polar and Marine Research
Telegrafenberg A45
14473 Potsdam

Reviewers' Comments to Author and Author's Answers to Reviewers:

Reviewer: 1

Comments to the Author

**Report #1 Submitted on 24 Jun 2025 Referee #1: Eryuan Liang, liangey@itpcas.ac.cn**

RC1: The authors addressed most of my previous comments. Based on revised manuscript, I have following minor comments.

L235-249: It is better to provide references for the numerical figures. Alternatively, show indicate your analysis, if these results are from your analysis.

AC: We have revised the text in the methods section to clarify the processes implemented in the snow module and the principles on which they are based.

RC1: As figure 3 and 4 give clear information on study sites, thus, Table 1 could be moved to supplementary information

AC: We respectfully disagree with the referee's view that Figures 3 and 4 convey the same information as Table 1, and therefore wish to retain Table 1.

RC1: Figure 5, 6 and 7 could be merged into a single figure. It would be helpful to readers see the site-specific features at a glance

AC: In line with the referee's suggestion, we have combined Figures 5 to 7 into a single figure and renumbered the subsequent figures accordingly.

RC1: Table 5: most of information repeated from earlier figures and tables and could be moved to supplementary information.

AC: We believe that Table 5, in addition to the preceding tables containing numerous numerical values, offers a clear overview of the results and the extent to which expectations were confirmed. For this reason, and contrary to the referee's suggestion, we would prefer not to move this table to the supplementary information.

RC1: Several references with missing volume, issue and page, or article numbers, please check them carefully.

AC: Where available, we have supplemented the missing information in the references. Some older articles do not have a DOI.

**Report #2 Submitted on 11 Aug 2025**

RC2: The reviewed version of the manuscript has improved its quality and clarity. The authors also replied accurately to the comments I raised at the first revision stage. I appreciate the fact the authors supported their replies with extensive references. Regarding the general comment raised at the first revision stage I have no major observations at the authors' replies.

Regarding the specific comments, I appreciate the authors restructured the introduction section by shortening it and by including a figure representing all the processes/factors affecting alpine treeline move. For the comments involving the methodological section the inclusion of figure 2 and the different specifics regarding the model workflow really improved the comprehension and readability of the case study. Maps of the areas and formulas are included as suggested.

**Best regards**

In the end I would say the authors have done a great job and in my opinion the manuscript can be considered for publication.

AC: Referee 2 had no additional requests for changes.

---

## Author Response (AR3)

**Subject: Cover Letter for 2nd Resubmission of Manuscript**

Dear Prof. Dr. Garbarino, Dr. Hagedorn, Dr. Migliavacca and reviewers,

I am pleased to submit the revised manuscript, entitled 'The significant role of snow in shaping alpine treeline responses in modelled boreal forests' for your careful review and consideration for publication in your special issue on the interactions between abiotic and biotic factors shaping vegetation patterns in treeline ecotones.

I wish to express my sincere gratitude for your continued feedback and constructive comments during the review of the revised version of our manuscript. We have carefully addressed each of the reviewers' comments and recommendations, which has further improved the quality of our study. I am confident that the revised manuscript is now even more in line with the high standards set by *Biogeoscience*.

We are looking forward to your evaluation and the possibility of having the revised manuscript accepted for publication in *Biogeoscience*. Please feel free to contact me should you require any further clarification or information.

Thank you for your time and consideration.

Sincerely, on behalf of all authors, Sarah Haupt

Polar Terrestrial Environmental Systems Alfred-Wegener-Institute, Helmholtz Centre for Polar and Marine Research Telegrafenberg A45 14473 Potsdam

Editor's Comments to Author and Author's Answers to Editor:

**16 Sep 2025**

**Associate editor decision: Publish subject to minor revisions (review by editor)**

Editor: One aspects it is still unclear to me is the extremely high tree density presented in the supplementary materials B1, and in Figure 7. I am not an expert of treelines, but from a brief literature review it seems that the values presented are rather high and I cannot find in the manuscript a comment on tha tree density obtained by the model. Can you please comment on this point, provide references and a clarification on this aspect?

AC: Thank you very much for your observation. The figures in Figure 7 are correct, as tree density is presented here as a dimensionless value, with logarithmic transformation applied to improve data visualisation. You were right to note a conversion error in Figure B1; this has now been corrected, and the stem count values are accurate.

Editor: Other minor changes required: Figure 2 caption: please spell LAVESI

AC: As recommended, LAVESI has now been written out in full in the figure caption.

Editor: Figure 4: Please increase the label size of the legend (DEM, Slope, and TWI and associated unit).

AC: In accordance with your recommendation, the label font size in Figure 4 has been increased.

Editor: Please change the data and code availability statement to include the final Zenodo item

AC: All materials have now been uploaded to Zenodo, and the code availability statement has been revised accordingly.